# Application of fused graphical lasso to statistical inference for multiple sparse precision matrices

Qiuyan Zhang[1]*, Lingrui Li[1], Hu Yang[2]

**1** School of Statistics, Capital University of Economics and Business, Beijing, China, **2** School of Information, Central University of Finance and Economics, Beijing, China

* qyzhang@cueb.edu.cn

## Abstract

In this paper, the fused graphical lasso (FGL) method is used to estimate multiple precision matrices from multiple populations simultaneously. The lasso penalty in the FGL model is a restraint on sparsity of precision matrices, and a moderate penalty on the two precision matrices from distinct groups restrains the similar structure across multiple groups. In high-dimensional settings, an oracle inequality is provided for FGL estimators, which is necessary to establish the central limit law. We not only focus on point estimation of a precision matrix, but also work on hypothesis testing for a linear combination of the entries of multiple precision matrices. We apply a de-biasing technology, which is used to obtain a new consistent estimator with known distribution for implementing the statistical inference, and extend the statistical inference problem to multiple populations. The corresponding de-biasing FGL estimator and its asymptotic theory are provided. A simulation study and an application of the diffuse large B-cell lymphoma data show that the proposed test works well in high-dimensional situation.

## Introduction

Undirected graphical models are popular tools for representing the network structure of data and have been widely applied in many domains, such as machine learning, gene pattern recognition, and financial data analysis. Letting $\mathbf{x} = (\mathbf{x}_1, \ldots, \mathbf{x}^p)^T$ be a p-variate normal random vector with mean vector $\mu$ and covariance $\Sigma_0$ ($\Sigma_0$ is positive definite), the precision matrix (or concentration matrix) is denoted the inverse of the covariance matrix, i.e., $\Theta_0 := \Sigma_0^{-1}$. The graphical models capture conditional dependence relationships between random variables via non-zero entries in a precision matrix. If $\Theta_{0ij} \neq 0$, $\mathbf{x}^i$ and $\mathbf{x}^j$, $i, j = 1, \ldots, p$ are dependent on each other, given all other variables. Meanwhile, the zero entries in the precision matrix correspond to pairs of variables that are conditionally independent given other variables. Therefore, the graph model is closely related to the precision matrix. The research of estimating and testing of a precision matrix have been a rapidly growing research direction in the past few years.

**Data Availability Statement:** All data come from website http://portals.broadinstitute.org/cgibin/cancer/datasets.cgi. However, we do not have the right to share this data.

**Funding:** the author Q.Y. Zhang is supported by Program for youth innovation Research in Capital University of Economics and Business (QNTD202207).

Letting $\mathbf{x}_1, \ldots, \mathbf{x}_n$ be a sequence of independent and identically distributed (i.i.d.) observations from the population $\mathbf{x}$, $\mathbf{X}_{p \times n} := (\mathbf{x}_1, \ldots, \mathbf{x}_n)$. A natural estimator of the precision matrix is the inverse of the sample covariance matrix $\hat{\Sigma}$, where $\hat{\Sigma} = \frac{1}{n}\mathbf{X}^T\mathbf{X}$. On the one hand, in high-dimensional settings, Johnstone [1] proposed that the eigenvalues of the sample covariance matrix do not converge to the corresponding eigenvalue of the population covariance matrix for $\Sigma = \mathbf{I}$. Consequently, this estimator becomes invalid when the dimension $p$ is comparable to the sample size $n$. On the other hand, the sample covariance matrix is singular in a $p > n - 1$ setting. This will produce non-negligible errors in using $\hat{\Sigma}_n^{-1}$ to estimate $\Theta_0$. In addition, a sparse (i.e., many entries are either zero or nearly so) assumption for a high-dimensional precision matrix is essential, since the zero entries imply the conditional independence structures, which are what we are most concerned with in the graphical model. In general, $\hat{\Sigma}_n^{-1}$ does not have a sparsity construction. How to estimate the sparse precision matrix in high-dimensional settings is an intractable problem.

In recent years, various proposals have been put forward for estimating a precision matrix in high-dimensional situations, among which the graphical model with sparsity-promoting penalties is valid for obtaining a sparse estimator. By applying the $l_1$ (lasso penalty) to the entries of the concentration matrix, Yuan and Lin [2] proposed a max-det algorithm to obtain the estimator of $\Theta_0$. The convergence result of the estimator is derived under a $p$ fixed assumption. Using a coordinate descent procedure, Friedman et al. [3] provided an algorithm for solving a graphical Lasso estimator that is remarkably fast, even if $p > n$. Rothman et al. [4] investigated a sparse permutation invariant covariance estimator, and established a convergence rate of the estimator in the Frobenius norm as both data dimension $p$ and sample size $n$ are allowed to grow, and showed that the rate explicitly depends on how sparse the true concentration matrix is. For additional theoretical details on penalized likelihood methods for graphical models, see Fan et al. [5], Ravikumar et al. [6], Xue and Zou [7], and Yuan et al. [8].

The above-mentioned methods focus on estimating a single graphical model. However, joint estimators perform better in recovering the truth graphs for multiple graphical models, when graphs sharing the similar structure. Guo et al. [9] studied joint estimation of precision matrices that have a hierarchical structure assumption. Zhang et al. [10] proposed a new joint group lasso penalty to restore the joint graphical model. Their method was applied for multiple gene networks data with several subpopulations and data types. A fused graphical lasso was proposed by Danaher et al. [11] with a penalty imposing a similar structure of a precision matrix across groups. Supposing that $\mathbf{X}_{p \times n_k}^{[k]} := (\mathbf{x}_1^{[k]}, \ldots, \mathbf{x}_{n_k}^{[k]})$ are sample matrices, and $\mathbf{x}_i^{[k]} \in R^p (i = 1, \ldots, n_k)$ are sampled i.i.d. from a distribution with mean $\mu^{[k]}$ and covariance $\Sigma_0^{[k]}$, for $k = 1, \ldots, K$, we assume $\mu^{[k]} = 0$ without loss of generality. To simplify notation, we omit the subscript of $\mathbf{X}_{p \times n_k}^{[k]}$, and denote the sample matrices as $\mathbf{X}^{[k]}$. The population precision matrix is defined as the inverse of the population covariance matrix, i.e., $\Theta_0^{[k]} = (\Sigma_0^{[k]})^{-1}$. The estimators of precision matrices $\{\Theta_0^{[k]}\}$ are investigated by minimizing the negative penalized log likelihood

$$\{\hat{\Theta}^{[k]}\} = \arg\min_{\{\Theta^{[k]} \in \mathcal{S}^{++}\}} \sum_k \{tr(\hat{\Sigma}^{[k]}\Theta^{[k]}) - \log\det(\Theta^{[k]})\} + \mathbf{P}(\{\Theta^{[k]}\}), \tag{1}$$

where $\mathbf{P}(\{\Theta^{[k]}\})$ denotes the penalty function, the $\{\hat{\Theta}^{[k]}\}$ are the minimizers of (1), and we optimize over the symmetric positive-definite matrices set $\mathcal{S}^{++}$. The fused graphical lasso (FGL) is

the solution to optimization problem (1) with the fused lasso penalty

$$\mathbf{P}(\{\Theta^{[k]}\}) = \lambda \sum_{k=1}^{K} ||(\Theta^{[k]})^-||_1 + \rho \sum_{k<k'} ||(\Theta^{[k]} - \Theta^{[k']})^-||_1, \tag{2}$$

where $\lambda$ and $\rho$ are non-negative regularization parameters, $(\Theta^{[k]})^-$ represents the matrix obtained by setting the diagonal elements of $(\Theta^{[k]})$ to zero, and $|| \cdot ||_1$ denotes the $l_1$ norm of a vector or matrix. It is reasonable to restrict non-diagonal elements of $\Theta^{[k]}$, since we are most concerned with the conditional independence cross-different variables. Note that the first term in (2) is the classical lasso penalty, which shrinks the coefficients toward 0 as $\lambda$ increases. It guarantees discovery of the sparse estimators $\{\hat{\Theta}^{[k]}\}$ of the model. The penalty on $(\Theta^{[k]} - \Theta^{[k']})^-$ indicates that the elements of $\hat{\Theta}^{[1]}, \ldots, \hat{\Theta}^{[K]}$ have a similar network structure across classes.

An approach for the estimation of the joint graphical models largely relies on penalized estimation. The penalty biases the estimates toward the assumed structure, which makes hypothesis tests for precision matrices more challenging. Work on statistical inference for low-dimensional parameters in graphical models has recently been carried out (Janková and van de Geer [12]; Janková and van de Geer [13]; Ren et al. [14]; Yu et al. [15]) based on the $l_1$-penalized estimator. Janková and van de Geer [12] provided a de-biasing technique to obtain a new consistent estimator with known distribution. However, these approaches were developed only in the setting in which the parameters of one graph are inferred. In contrast, studies of inference techniques using estimators obtained from cross-group penalization are much fewer. The work on statistical inference for multiple graphical models is an interesting area open for future research. Inspired by Janková and van de Geer [12], we not only give FGL estimators of multiple precision matrices from co-movement data, but also test the linear combination of the entries of these precision matrices. The core of the proposed method is based on the de-biasing technique, and we implement statistical inference of the precision matrices under high-dimensional settings according to the proposed central limit theorem.

The rest of this paper is organized as follows. In Main results section, we give the oracle inequality for multiple estimators with a FGL penalty and its weighted version. Testing the hypothesis for the linear combination of corresponding entries of multiple precision matrices is also considered in this section. Based on de-biasing technology, the CLT of the proposed statistics for multiple populations is also derived in this section. In Numerical study part, we report the results of simulations. In Real Data Application, we apply the proposed method to identification of gene-to-gene interaction of the diffuse large B-cell lymphoma data. All technical details are relegated to the Proof of Theorem part.

## Main results

We assume following notation throughout the paper. For a matrix $A = (a_{ij})_{i,j=1}^{p}$, we denote $(A)_{ij}$ as $(i, j)$-entry of $A$, or denote its $(i, j)$-entry as $A_{ij}$ to simplify the notation. We write $|A|$ for the determinant of $A$, and the trace of matrix $A$ is denoted $tr(A)$. Letting $A^+ = diag(A)$ for a diagonal matrix with the same diagonal as $A$, $A^- = A - A^+$. $||A||_F^2 = \sum_{i,j} a_{ij}^2$ denotes the Frobenius norm (also known as the matrix 2-norm). We use the notation $||A||_\infty = \max_{i,j} |a_{ij}|$ for the supremum norm of a matrix $A$, and $|||A|||_1 := \max_j \Sigma_i |a_{ij}|$ for the $l_1$-operator norm.

We write $f(n) = \mathcal{O}(g(n))$ if $f(n) \leq cg(n)$ for some constant $c < \infty$, and $f(n) = \Omega(g(n))$ if $f(n) \geq c'g(n)$ for some constant $c' > 0$. The notation $f(n) \asymp g(n)$ means that $f(n) = \mathcal{O}(g(n))$ and $f(n) = \Omega(g(n))$. In the common high-dimensional setting, the dimension $p$ is allowed to grow to infinity. The dimension is comparable, substantially larger or smaller than the sample

size. We set sample sizes $n_1 \asymp \ldots \asymp n_K \asymp n$ throughout the paper, and $n^* = n_1 + \ldots + n_K$ going to infinity. Furthermore, for notational simplicity, we assume that $n_1 = \ldots = n_K = n$.

## Oracle inequality

To obtain the oracle inequality of multiple estimators of FGL models, we introduce some notation related to the sparsity assumptions on the entries of the true precision matrix. Letting

$$S_k := \{(i,j) : \Theta_{0ij}^{[k]} \neq 0, i \neq j\},$$

where $\Theta_{0ij}^{[k]}$ is the $(i,j)$-entry of $\Theta_0^{[k]}$ and $s_k = |S_k|$ is the cardinality of $S_k$, we adopt the boundedness of the eigenvalues of the true precision matrix and certain tail conditions proposed by Janková and Van De Geer [12].

**Condition 1 (Bounded eigenvalues)** *There exist universal constants L for k such that*

$$0 < L < \Lambda_{\min}(\Theta_0^{[k]}) \leq \Lambda_{\max}(\Theta_0^{[k]}) < 1/L < \infty,$$

where $\Lambda_{\min}$ and $\Lambda_{\max}$ denote the minimum and maximum eigenvalues of a matrix, respectively.

**Condition 2 (Sub-Gaussianity vector condition)** *The observations $\mathbf{x}_i^{[k]}$, $i = 1, \ldots, n_k$, are uniformly sub-Gaussian vectors in the respective groups.*

We propose the oracle inequality for FGL lasso under the $K = 2$ situation.

**Theorem 1** *Supposing that Conditions 1 and 2 hold, for $k = 1, 2$, tuning parameter $\lambda$ satisfying $2(\rho + \lambda_0) \leq \lambda \leq c/8L$, and $\frac{8\lambda^2(s_1+s_2)}{c} + \frac{4p\lambda_0^2}{c} \leq \lambda_0/2L$. On the set $\{\max_k||\hat{\Sigma}^{[k]} - \Sigma_0^{[k]}||_\infty \leq \lambda_0\}$, $k = 1, 2$, it holds that*

$$c\sum_{k=1}^{2}||\hat{\Theta}^{[k]} - \Theta_0^{[k]}||_F^2 + \lambda\sum_{k=1}^{2}||(\hat{\Theta}^{[k]} - \Theta_0^{[k]})^-||_1 \leq \frac{8\lambda^2(s_1+s_2)}{c} + \frac{4p\lambda_0^2}{c},$$

*and*

$$\sum_{k=1}^{2}|||\hat{\Theta}^{[k]} - \Theta_0^{[k]}|||_1 \leq \frac{4\lambda(8s_1 + 8s_2 + p)}{c},$$

*where $c = 1/(8L^2)$.*

**Remark 1** *From the inequality, we must select $\lambda$ so that $\lambda p \to 0$ as $n \to \infty$ to ensure consistency, which is not satisfied by a sub-Gaussianity random vector. Thus, the condition $\lambda p \to 0$ excludes the $p \gg n$ situation.*

The FGL does not take into account that the variables have, in general, different scaling. Thus, we consider the weighted FGL. The minimizer of the optimization problem (1) with weighted FGL penalty

$$\mathbf{P}(\{\Theta^{[k]}\}) = \lambda\sum_{k}\sum_{i\neq j} \hat{W}_{ii}^{[k]}\hat{W}_{jj}^{[k]}|\Theta_{ij}^{[k]}| + \rho\sum_{k<k'}\sum_{i\neq j}|\hat{W}_{ii}^{[k]}\hat{W}_{jj}^{[k]}\Theta_{ij}^{[k]} - \hat{W}_{ii}^{[k']}\hat{W}_{jj}^{[k']}\Theta_{ij}^{[k']}| \quad (3)$$

is denoted $\{\hat{\Theta}_w^{[k]}\}$, where $\hat{W}^{[k]} = \left[diag(\hat{\Sigma}^{[k]})\right]^{\frac{1}{2}}$. Further, the population correlation matrix is denoted $R_0^{[k]}$ and the sample correlation matrix is denoted

$$\hat{R}^{[k]} = (\hat{W}^{[k]})^{-1}\hat{\Sigma}^{[k]}(\hat{W}^{[k]})^{-1}.$$

If we substitute $\hat{R}^{[k]}$ for $\hat{\Sigma}^{[k]}$, the minimizer of

$$\arg\min_{\{\Theta^{[k]} \in \mathcal{S}^{++}\}} \sum_k \{tr(\hat{R}^{[k]}\Theta^{[k]}) - \log\det(\Theta^{[k]})\} + \mathbf{P}(\{\Theta^{[k]}\}) \tag{4}$$

with a FGL penalty (2) is denoted $\{\hat{\Theta}_R^{[k]}\}$, which is a matter of estimating the parameter by the normalized data. Then,

$$\hat{\Theta}_R^{[k]} = \hat{W}^{[k]}\hat{\Theta}_w^{[k]}\hat{W}^{[k]},$$

which means, essentially, that $\hat{\Theta}_R^{[k]}$ are the estimators of $\Theta_{R0}^{[k]} := (R_0^{[k]})^{-1}$.

**Theorem 2** *Under the conditions of Theorem 1, on the set* $\{\max_k ||\hat{R}^{[k]} - R_0^{[k]}||_\infty \le \lambda_0\}$, $k = 1$, 2, *it holds that*

$$c\sum_{k=1}^{2}||\hat{\Theta}_R^{[k]} - \Theta_{R0}^{[k]}||_F^2 + \lambda\sum_{k=1}^{2}||(\hat{\Theta}_R^{[k]} - \Theta_{R0}^{[k]})^-||_1 \le \frac{8\lambda^2(s_1 + s_2)}{c}, \tag{5}$$

$$\sum_{k=1}^{2}|||\hat{\Theta}_R^{[k]} - \Theta_{R0}^{[k]}|||_1 \le \frac{32\lambda(s_1 + s_2)}{c}, \tag{6}$$

*and*

$$\sum_{k=1}^{2}|||\hat{\Theta}_w^{[k]} - \Theta_0^{[k]}|||_1 \le \frac{32\lambda(s_1 + s_2)}{c}. \tag{7}$$

It is natural to extend this conclusion to the $K > 2$ FGL model. For $k = 1, \ldots, K$ and the $K > 2$ situation, we obtain the following theorem.

**Theorem 3 (Multiple FGL model)** *Supposing that Conditions 1 and 2 hold, for $K > 2$,* $2\left(\frac{K(K-1)}{2}\rho + \lambda_0\right) \le \lambda \le c/8L$, *and* $\frac{8\lambda^2\sum_{k=1}^{K}s_k}{c} + \frac{2Kp\lambda_0^2}{c} \le \lambda_0/2L$, *on the set* $\{\max_k ||\hat{\Sigma}^{[k]} - \Sigma_0^{[k]}||_\infty \le \lambda_0\}$, $k = 1, \ldots, K$, *it holds that*

$$c\sum_{k=1}^{K}||\hat{\Theta}^{[k]} - \Theta_0^{[k]}||_F^2 + \lambda\sum_{k=1}^{K}||(\hat{\Theta}^{[k]} - \Theta_0^{[k]})^-||_1 \le \frac{8\lambda^2\sum_{k=1}^{K}s_k}{c} + \frac{2Kp\lambda_0^2}{c} \tag{8}$$

*and*

$$\sum_{k=1}^{K}|||\hat{\Theta}^{[k]} - \Theta_0^{[k]}|||_1 \le \frac{2K\lambda\left(8\sum_{k=1}^{K}s_k + \frac{Kp}{2}\right)}{c}. \tag{9}$$

**Theorem 4 (Multiple FGL model for weighted version)** *Under the conditions of Theorem 3, on the set* $\{\max_k ||\hat{R}^{[k]} - R_0^{[k]}||_\infty \le \lambda_0\}$, $k = 1$, 2, *it holds that*

$$c\sum_{k=1}^{K}||\hat{\Theta}_R^{[k]} - \Theta_{R0}^{[k]}||_F^2 + \lambda\sum_{k=1}^{K}||(\hat{\Theta}_R^{[k]} - \Theta_{R0}^{[k]})^-||_1 \le \frac{8\lambda^2\sum_{k=1}^{K}s_k}{c}, \tag{10}$$

$$\sum_{k=1}^{K}|||\hat{\Theta}_R^{[k]} - \Theta_{R0}^{[k]}|||_1 \le \frac{16K\lambda\sum_{k=1}^{K}s_k}{c}, \tag{11}$$

*and*

$$\sum_{k=1}^{K} |||\hat{\Theta}_w^{[k]} - \Theta_0^{[k]}|||_1 \frac{16K\lambda \sum_{k=1}^{K} s_k}{c}. \tag{12}$$

## Asymptotic property

We not only focus on the point estimation of multiple precision matrices, but also on hypothesis testing for the linear combination of the entries of the precision matrices over two groups. One may want to test whether the elements of the precision matrix over two groups are equal:

$$H_0 : \Theta_{0ij}^{[1]} = \Theta_{0ij}^{[2]} \quad vs. \quad H_1 : \Theta_{0ij}^{[1]} \neq \Theta_{0ij}^{[2]}. \tag{13}$$

To test Hypothesis (13), we aim to obtain confidence intervals for estimators based on the de-biasing technique, which is imposed for eliminating the bias associated with the penalty. The de-biasing estimator is defined as $\hat{\Theta}_d^{[k]} = 2\hat{\Theta}^{[k]} - \hat{\Theta}^{[k]}\hat{\Sigma}^{[k]}\hat{\Theta}^{[k]}$. The difference between the de-biasing estimator and the true value can be decomposed into two parts as follows:

$$\hat{\Theta}_d^{[k]} - \Theta_0^{[k]} = \Xi^{[k]} + \Upsilon^{[k]},$$

where

$$\Xi^{[k]} = -\Theta_0^{[k]}(\hat{\Sigma}^{[k]} - \Sigma_0^{[k]})\Theta_0^{[k]},$$

$$\Upsilon^{[k]} = -(\hat{\Theta}^{[k]} - \Theta_0^{[k]})(\hat{\Sigma}^{[k]} - \Sigma_0^{[k]})\Theta_0^{[k]} - (\hat{\Theta}^{[k]} - \Theta_0^{[k]})(\hat{\Sigma}^{[k]}\hat{\Theta}^{[k]} - \mathbf{I}_p).$$

Under the compatibility conditions, Janková and van de Geer [16] proposed that the $(i, j)$-entry of $\hat{\Theta}_d^{[k]} - \Theta_0^{[k]}$ has an asymptotic normality property, and $\sqrt{n}||\Upsilon^{[k]}||_\infty$ converges to zero in probability. Thus, for testing Hypothesis (13), we construct the testing statistic

$$T_{ij} := \left( \hat{\Theta}_d^{[1]} - \hat{\Theta}_d^{[2]} \right)_{ij} = \left[ 2\hat{\Theta}^{[1]} - \hat{\Theta}^{[1]}\hat{\Sigma}^{[1]}\hat{\Theta}^{[1]} - (2\hat{\Theta}^{[2]} - \hat{\Theta}^{[2]}\hat{\Sigma}^{[2]}\hat{\Theta}^{[2]}) \right]_{ij} \tag{14}$$

using de-biasing estimators.

For $K = 2$, we let

$$s = \max\{s_1, s_2\}, \quad d = \max\{d_1, d_2\},$$

where

$$d_k = \max_{j=1,\dots,p} |D_j^{[k]}|, \quad D_j^{[k]} = \{(i,j) : \Theta_{0ij}^{[k]} \neq 0, i \neq j\}.$$

Next, we establish the central limit theorem for $T_{ij}$.

**Theorem 5** *Assuming Conditions 1, 2, and $\lambda \asymp \rho \asymp \sqrt{\log p/n}$ and $(p + s)\sqrt{d} = o(\sqrt{n}/\log p)$, it holds that*

$$\hat{\Theta}_d^{[1]} - \hat{\Theta}_d^{[2]} - (\Theta_0^{[1]} - \Theta_0^{[2]}) = \Xi^{[1]} - \Xi^{[2]} + rem, \tag{15}$$

*where*

$$||rem||_\infty = ||\Upsilon^{[1]} - \Upsilon^{[2]}||_\infty = o_p(1/\sqrt{n}), \tag{16}$$

and $o_p$ denotes the convergence in probability. Moreover,

$$\sqrt{n}[T_{ij} - \mathbf{\Theta}_{0ij}] \to_D N(0, \sigma_{ij}^2), \tag{17}$$

where $\mathbf{\Theta}_{0ij} = (\Theta_0^{[1]} - \Theta_0^{[2]})_{ij}$.

To complete the testing procedure, we use the consistent estimator $\hat{\sigma}_{ij}^2 = (\hat{\mathbf{\Theta}}^{[1]})_{ii}(\hat{\mathbf{\Theta}}^{[1]})_{jj} + (\hat{\mathbf{\Theta}}^{[1]})_{ij}^2 + (\hat{\mathbf{\Theta}}^{[2]})_{ii}(\hat{\mathbf{\Theta}}^{[2]})_{jj} + (\hat{\mathbf{\Theta}}^{[2]})_{ij}^2$ for Theorem 5. Theorem 5 provide a practical and efficient way of obtaining the p value and critical value for the test statistic. Under a null hypothesis, we observe that $\Theta_{0ij}^{[1]} - \Theta_{0ij}^{[2]} = 0$. For an $\alpha$ level of significance, we reject $H_0$ if $|\sqrt{n}T_{ij}/\hat{\sigma}_{ij}^2| > \xi_{\alpha/2}$, where $\xi_\alpha$ is the $1 - \alpha$ upper quantile of the standard normal distribution.

Theorem 5 requires a stronger sparsity condition than the corresponding oracle-type inequality in Theorem 1. According to the convergence rate of $(p + s)\sqrt{d}$, Theorem 5 applies to the $p \ll n$ situation. For $p \gg n$, we provide the following theorem.

**Theorem 6** *Assuming Conditions 1, 2, and $\lambda \asymp \rho \asymp \sqrt{\log p/n}$ and $s\sqrt{d} = o(\sqrt{n}/\log p)$, for the $p \ll n$ regime, the* Eq (22) *holds with $\hat{\mathbf{\Theta}}_w^{[k]}$, where*

$$||rem||_\infty = o_p(1/\sqrt{n}). \tag{18}$$

*In addition,*

$$\sqrt{n}[T_{wij} - \mathbf{\Theta}_{0ij}] \to_D N(0, \sigma_{ij}^2), \tag{19}$$

where $T_{wij} = (2\hat{\mathbf{\Theta}}_w^{[1]} - \hat{\mathbf{\Theta}}_w^{[1]}\hat{\Sigma}^{[1]}\hat{\mathbf{\Theta}}_w^{[1]})_{ij} - (2\hat{\mathbf{\Theta}}_w^{[2]} - \hat{\mathbf{\Theta}}_w^{[2]}\hat{\Sigma}^{[2]}\hat{\mathbf{\Theta}}_w^{[2]})_{ij}$.

We do not need to impose the so-called irrepresentability condition on $\Sigma$ to derive the theoretical properties of our estimators, in contrast to Brownlees et al. [17].

In addition, for the multi-sample precision matrix hypothesis problem, one may want to test a linear hypothesis testing problem:

$$H_0 : a_1\Theta_{0ij}^{[1]} + \ldots + a_K\Theta_{0ij}^{[K]} = 0 \quad vs. \quad H_1 : \text{not} \quad H_0, \tag{20}$$

where $a_1, \ldots, a_K$ are known constants. Similar to the two-sample case, we proposed the test statistic

$$a_1\hat{\mathbf{\Theta}}_{dij}^{[1]} + \ldots + a_K\hat{\mathbf{\Theta}}_{dij}^{[K]}. \tag{21}$$

For the $K > 2$ multiple situation, we assume $s = \max\{s_1, \ldots, s_K\}$ and $d = \max\{d_1, \ldots, d_K\}$. Consequently, we establish the asymptotic normality of the proposed statistic in the following corollary, i.e., Corollary 1.

**Corollary 1** *Under the assumptions of Theorem 5, it holds that*

$$f(\hat{\mathbf{\Theta}}_d^{[1]}, \ldots, \hat{\mathbf{\Theta}}_d^{[K]}) - f(\Theta_0^{[1]}, \ldots, \Theta_0^{[K]}) = f(\Xi^{[1]}, \ldots, \Xi^{[K]}) + rem, \tag{22}$$

$$||rem||_\infty = ||f(\Upsilon^{[1]}, \ldots, \Upsilon^{[K]})||_\infty = o_p(1/\sqrt{n}), \tag{23}$$

where $f(x_1, \ldots, x_K) = a_1x_1 + \ldots + a_Kx_K$. In addition,

$$\sqrt{n}[T_{ij} - \mathbf{\Theta}_{0ij}] \to_D N(0, \sigma_{ij}^2), \tag{24}$$

where $T_{ij} = f\left(\hat{\mathbf{\Theta}}_{dij}^{[1]}, \ldots, \hat{\mathbf{\Theta}}_{dij}^{[K]}\right)$ and $\mathbf{\Theta}_{0ij} = f\left(\Theta_{0ij}^{[1]}, \ldots, \Theta_{0ij}^{[K]}\right)$.

The asymptotic variance $\sigma_{ij}$ in Corollary 1 is unknown, so to construct confidence intervals we use a consistent estimator

$$\hat{\sigma}_{ij}^2 = f_v([(\hat{\Theta}^{[1]})_{ii}(\hat{\Theta}^{[1]})_{jj} + (\hat{\Theta}^{[1]})_{ij}^2], \ldots, [(\hat{\Theta}^{[K]})_{ii}(\hat{\Theta}^{[K]})_{jj} + (\hat{\Theta}^{[K]})_{ij}^2]),$$

where $f_v(x_1, \ldots, x_K) = a_1^2 x_1 + \ldots + a_K^2 x_K$. In addition, a weighted version is proposed as follows.

**Corollary 2** *Under the assumptions of Theorem 6, the residual term in* (23) *converges in probability with rate* $1/\sqrt{n}$, *and CLT in* (24) *holds by replacing* $\hat{\Theta}^{[k]}$ *by* $\hat{\Theta}_w^{[k]}$, *which is obtained by solving the weighted FGL optimization problem.*

## Numerical study

Simulation experiments were carried out to evaluate the performance of the proposed de-biasing FGL test. We considered the sparse graphical model, and a random sample was generated from the multivariate normal distribution $N(0_p, (\Theta_0^{[k]})^{-1})$ with a population covariance matrix defined as the inverse of the population precision matrix.

To solve the graphical lasso problem with a certain penalty, we refer to the alternating direction method of multiplier (ADMM) algorithm, since it is guaranteed to converge to the global optimum. For more details, the reader is referred to Boyd et al. [18] and Danaher et al. [11]. When an objective method for selecting tuning parameters $\lambda$ and $\rho$ is required, the approximations of the Akaike information criterion (AIC), Bayesian information criterion, or cross-validation method can be used to select tuning parameters. The AIC method was chosen for the following simulation, and $\lambda$ and $\rho$ both range from 0.05 to 0.3 with a step of 0.0086, where the step is derived by $(0.3 - 0.05)/(30 - 1)$.

In addition, all the reported simulation results are based on 500 simulations with a nominal significance level of 0.05, and we set the dimension to 100.

### Fluctuations of test

We illustrated the theoretical asymptotic normality result on simulated data for testing the two-sample problem (13), and we set precision matrices equal under a null hypothesis, i.e., $\Theta_0^{[1]} = \Theta_0^{[2]}$.

Letting $G$ be a $p \times p$ symmetric graph matrix with diagonal entries 0 and $\tilde{\alpha}$ percent of off-diagonal elements 1, and $U$ be $p \times p$ matrix with elements i.i.d. generated from the uniformly distribution on the interval (0, 1), i.e., $U(0, 1)$, we denote the elements of the symmetric matrix $\tilde{\Theta}$ as $\tilde{\theta}_{ij}$. For $i > j$,

$$\tilde{\theta}_{ij} = \frac{g_{ij}u_{ij} + g_{ji}u_{ji}}{2} - \mathbf{1}_{\{\frac{g_{ij}u_{ij} + g_{ji}u_{ji}}{2} < 0.5\}}, \tag{25}$$

where $g_{ij}$ and $u_{ij}$ are the $(i, j)$-entry of $G$ and $U$, respectively, and $\mathbf{1}_{\{\cdot\}}$ is the indicator function. For $i < j$, we set $\tilde{\theta}_{ij} = \tilde{\theta}_{ji}$. The diagonal entries of matrix $\tilde{\Theta}$ are zeros. Then, the precision matrix is generated as

$$\Theta_0^{[k]} = \tilde{\Theta} + \left(|\Lambda_{\min}(\tilde{\Theta})| + 0.1\right)\mathbf{I}_p. \tag{26}$$

This shows that the matrix generated is symmetric and positive definite. To make the non-zero entries go away from 0 and to generate a sparse matrix, we subtract 1 from the non-zero elements. In addition, the precision matrix generation procedure shows that $\tilde{\alpha}$ is a parameter controlling the sparsity. When $\tilde{\alpha} = 1$, a dense matrix is generated. As is well known, the

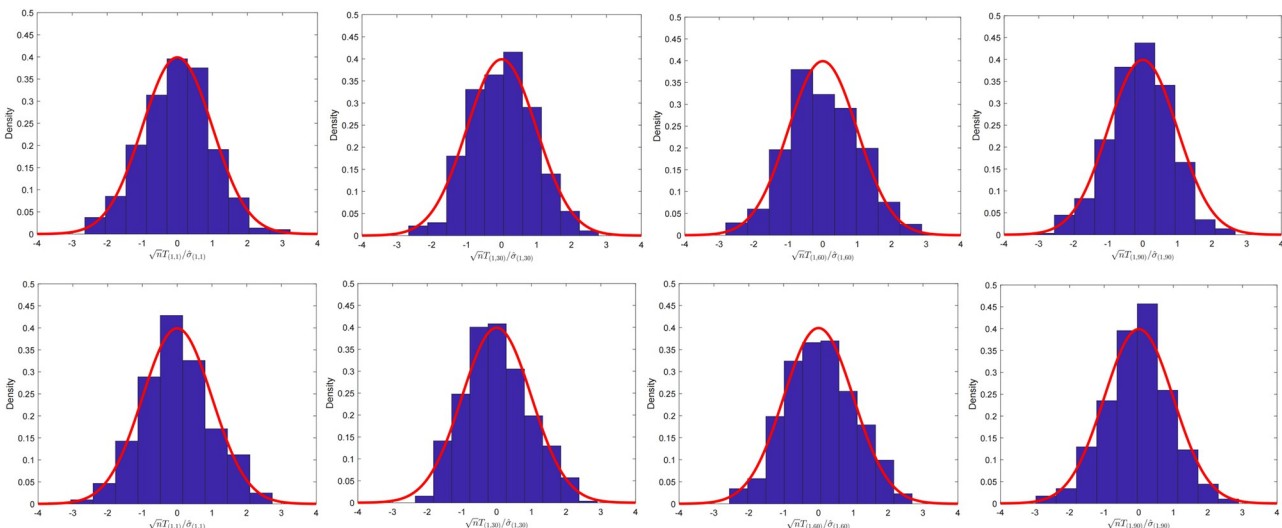

**Fig 1. The fluctuation for two-sample case with sparse precision matrix.** Histogram of $\sqrt{n}T_{ij}/\hat{\sigma}_{ij}$ for $\tilde{\alpha} = 0.01$. Here, $T_{(i,j)} = T_{ij}$ and $\hat{\sigma}_{(i,j)} = \hat{\sigma}_{ij}$. The setting is $(p, n) = (100, 200)$ with $(i, j) \in \{(1, 1), (1, 30), (1, 60), (1, 90)\}$ for four graphs in the first line. The sample size and dimension were set as $(p, n) = (100, 400)$ for four graphs in the second line.

sparsity of a matrix not only requires a small quantity of non-zero elements, but also a large absolute value of non-zero elements. The parameter $\tilde{\alpha}$ controls sparsity in terms of the number of sparse elements.

We examined the fluctuation of $\sqrt{n}T_{ij}/\hat{\sigma}_{ij}$ under $(p, n) = (100, 200)$ and $(p, n) = (100, 400)$ settings for the extremely sparse and dense precision matrix cases, respectively. For the extremely sparse precision matrix case, we set the parameter $\tilde{\alpha} = 0.01$, and for dense case we use $\tilde{\alpha} = 1$.

We simulated the fluctuation for the extremely sparse case as shown in Fig 1 and the dense case in Fig 2. The index $(i, j)$ in the simulation was intermittently chosen. In fact, the CLT provides the method for testing any element of the linear combination of the precision matrix. Theoretically, we can test for any index $(i, j)$-entry of $\mathbf{\Theta}_0$ whether the true value is zero or not.

### Average coverage probabilities

We demonstrate the performance of the test method for the $K = 2$ situation on testing the hypothesis as follows.

- **Equal Null**. Testing hypothesis (13);

- **Linear Null**. Testing the linear null hypothesis $H_0 : a_1 \Theta_{0ij}^{[1]} + a_2 \Theta_{0ij}^{[2]} = 0$, i.e.,

  $H_0 : \Theta_{0ij}^{[2]} = -\frac{a_1}{a_2} \Theta_{0ij}^{[1]}$. Without loss generation, we chose $-\frac{a_1}{a_2} = 0.5$ and $\Theta_{0ij}^{[1]}$ generated from (26).

From the global perspective, we used the average coverage, which is also considered in Janková and van de Geer [12]. Letting

$$I_{ij} := \left[ T_{ij} - 1.96 \frac{\sigma_{ij}}{\sqrt{n}}, T_{ij} + 1.96 \frac{\sigma_{ij}}{\sqrt{n}} \right] \tag{27}$$

be the 95% asymptotic confidence interval for $\mathbf{\Theta}_{0ij}$, we substitute the estimator $\hat{\sigma}_{ij}$ for $\sigma_{ij}$ to

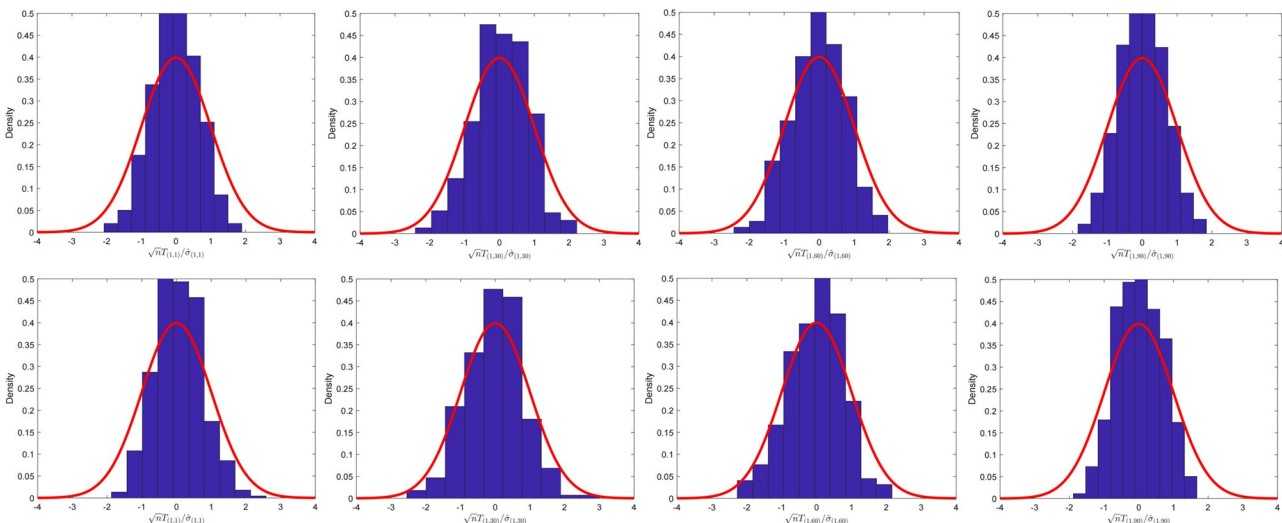

**Fig 2. The fluctuation for two-sample case with dense precision matrix.** Histogram of $\sqrt{n}T_{ij}/\hat{\sigma}_{ij}$ for $\tilde{\alpha} = 1$. Here, $T_{(i,j)} = T_{ij}$ and $\hat{\sigma}_{(i,j)} = \hat{\sigma}_{ij}$. The setting is $(p, n) = (100, 200)$ with $(i, j) \in \{(1, 1), (1, 30), (1, 60), (1, 90)\}$ for four graphs in the first line. The sample size and dimension were set to $(p, n) = (100, 400)$ for four graphs in the second line.

obtain the empirical version. The frequency of the true value being covered by the confidence interval (27) is defined as $\hat{\vartheta}_{ij}$. Then, the average coverage over a set $A$ is denoted

$$Avgcov_A = \frac{1}{|A|} \sum_{(i,j) \in A} \hat{\vartheta}_{ij}. \tag{28}$$

$S$ denotes the set of non-zero entries of $\Theta_{0ij}^{[1]}$. It is easy to check that $S = S_1 = S_2$ for the reason that $\Theta_{0ij}^{[1]}$ and $\Theta_{0ij}^{[2]}$ have same structure of sparsity for the Equal Null and Linear Null cases. Thus, for the different null hypotheses, we simulated the average coverage over $S$ and its complementary set $S^c$. The parameter of sparsity is $\tilde{\alpha} = 0.1, 0.5$, and $0.9$.

Partial results in Table 1 meet our expectation. However, we do not deny that the simulations are affected by randomness. In addition, the proposed method is based on the combination of estimation and hypothesis testing, which accumulates error. The simulation results provide guidance for practice.

**Table 1. Estimated average coverage probabilities for K = 2 situation.**

| 2*$\tilde{\alpha}$ | 2*$n$ | Equal Null | | Linear Null | |
|---|---|---|---|---|---|
| | | $S$ | $S^c$ | $S$ | $S^c$ |
| 2*0.1 | 200 | 0.9886 | 0.9875 | 0.9101 | 0.9824 |
| | 400 | 0.9885 | 0.9867 | 0.8607 | 0.9762 |
| 2*0.5 | 200 | 0.9880 | 0.9878 | 0.9384 | 0.9745 |
| | 400 | 0.9870 | 0.9868 | 0.8820 | 0.9647 |
| 2*0.9 | 200 | 0.9901 | 0.9899 | 0.9509 | 0.9751 |
| | 400 | 0.9889 | 0.9890 | 0.9091 | 0.9639 |

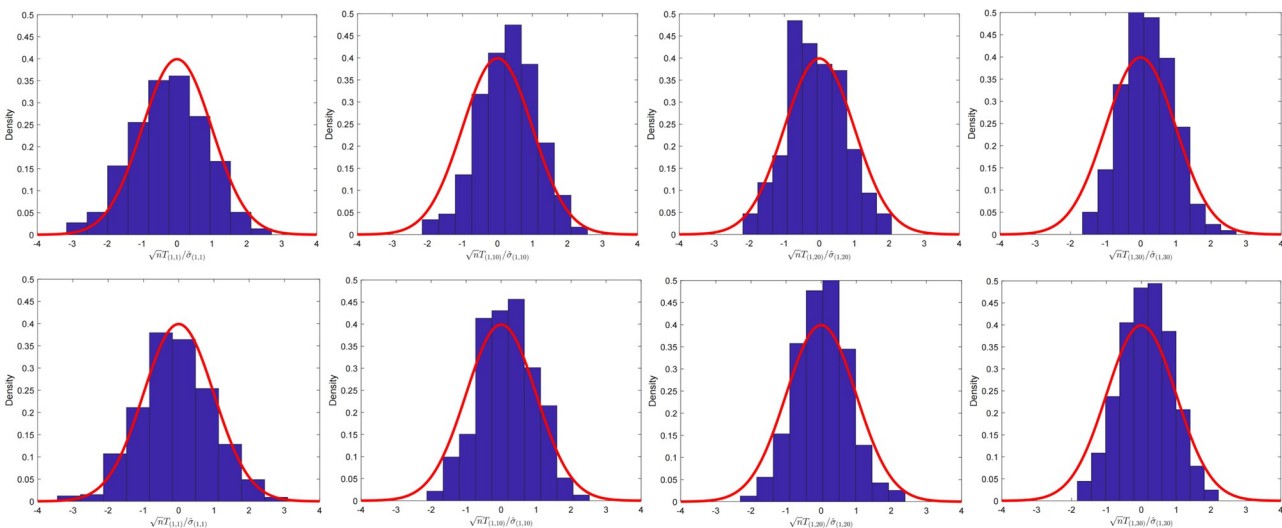

**Fig 3. The fluctuation for multiple-sample case with dense precision matrix.** Histogram of $\sqrt{n}T_{ij}/\hat{\sigma}_{ij}$ for $\tilde{\alpha} = 1$. Here, $T_{(i,j)} = T_{ij}$ and $\hat{\sigma}_{(i,j)} = \hat{\sigma}_{ij}$. The setting is $(p, n) = (100, 200)$ with $(i, j) \in \{(1, 1), (1, 10), (1, 20), (1, 30)\}$ for four graphs in the first line. The sample size and dimension were set to $(p, n) = (100, 400)$ for four graphs in the second line.

## Multiple FGL case

For the multiple FGL case, we examined the fluctuation of the statistic $T_{ij}$ for the $K = 3$ situation on testing the hypothesis as follows.

- **Three-sample Linear Null**. Testing hypothesis $H_0 : \Theta_{0ij}^{[3]} = -\frac{a_1}{a_3}\Theta_{0ij}^{[1]} - \frac{a_2}{a_3}\Theta_{0ij}^{[2]}$, where $-\frac{a_1}{a_3} = 0.6$ and $-\frac{a_2}{a_3} = 0.9$ are both generated from $U(0, 1)$. $\Theta_{0ij}^{[1]}$ and $\Theta_{0ij}^{[2]}$ are both generated from (26) with parameters 0.01 and 0.1, respectively.

  We set $-\frac{a_1}{a_3}$ and $-\frac{a_2}{a_3}$ to positive numbers, since the setting of hypothesis testing should guarantee that $\{\Theta_{0ij}^{[k]}\}_{k=1}^{3}$ are symmetric positive-definite matrices. Besides, for Three-sample Linear Null, $S$ denotes the set of non-zero entries of $\Theta_{0ij}^{[1]} + \frac{a_2}{a_3}\Theta_{0ij}^{[2]}$. The dimension and sample size are $(p, n) = (100, 200)$ and $(p, n) = (100, 400)$, respectively. Histograms of the proposed statistic $T_{ij}$ at the

$$(i, j) \in \{(1, 1), (1, 10), (1, 20), (1, 30)\}$$

locations of the precision matrix are presented in Fig 3.

## Real data application

The lymphoma is a malignant tumor with increasing incidence and mortality year by year. In this part, we apply the proposed method to two sets of diffuse large B-cell lymphoma (DLBCL) data, denoted DLBCL-A [19] and DLBCL-B [20], which is available at http://portals. broadinstitute.org/cgibin/cancer/datasets.cgi. Some brief information on these datasets can be found in Table 2. The DLBCL-A and DLBCL-B datasets have 3 subgroups, and the label and sample size of each subgroup are shown in the 5*th* column in Table 2. Both DLBCL-A dataset and DLBCL-B dataset have a high dimension with 662 genes but only a few observations with the sample size 141 for the DLBCL-A dataset and 180 for the DLBCL-B dataset.

**Table 2. Brief introduction to the gene profile expression datasets.**

| Dataset | $n$ | $p$ | Subgroups | Subgroup label (sample size) |
|---|---|---|---|---|
| DLBCL-A | 141 | 662 | 3 | I (49), II (50), III (42) |
| DLBCL-B | 180 | 662 | 3 | I (42), II (51), III (87) |

Typically, we test for differences in mean vectors across different disease subgroups, however, the role of gene-to-gene interactions in the data across different subtypes remains unclear. In this section, we use our test approach to identify whether the gene-to-gene interactions that controls lymphoma most behave the same across different disease subtypes. For distinct gene subtypes of the same disease gene data, we focus on testing the equality of two precision matrices. The hypothesis testing problem is

$$H_0 : \Theta_{0ij}^{type\ i} = \Theta_{0ij}^{type\ j} \quad vs. \quad H_1 : \Theta_{0ij}^{type\ i} \neq \Theta_{0ij}^{type\ j}$$

where *type i* and *type i* are chosen from I, II, III set in Table 2 and *type i ≠ type j*. We tune parameters with weighted FGL penalty in (3) by AIC criterion. After the tuning procedure, we estimate precision matrices, and then return a $p \times p$ matrix, whose $(i, j)$-th elements are p-value of statistic $T_{ij}$. The results are demonstrated in Figs 4 and 5.

As can be seen in the figure, the interactions between genes of DLBCL-A dataset are not the same among three different subtypes, while for DLBCL-B dataset, the interactions between genes of three different subtypes are mostly similar.

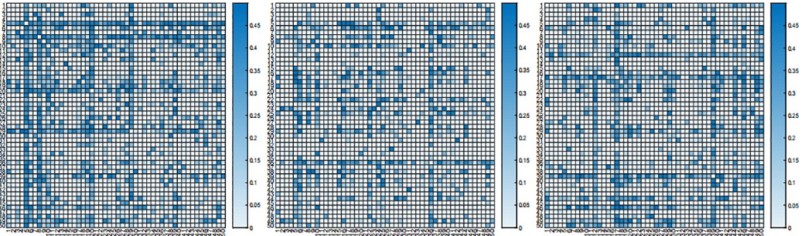

**Fig 4. The p-values of proposed test for DLBCL-A dataset.** P-values of $T_{ij}$ by comparing subtype I and subtype II (left), subtype II and subtype III (middle), and subtype I and subtype III (right) with DLBCL-A dataset.



**Fig 5. The p-values of proposed test for DLBCL-B dataset.** P-values of $T_{ij}$ by comparing subtype I and subtype II (left), subtype II and subtype III (middle), and subtype I and subtype III (right) with DLBCL-B dataset.

## Proof of theorem

### Proof of Theorem 1

To prove Theorem 1, we need a lemma of Janková and Van de Geer [16], which is present as follow.

**Lemma 7** *Let $f(\Delta) := tr(\Delta\Sigma_0) - [\log\det(\Delta + \Theta_0) - \log\det(\Theta_0)]$. Assume that $1/L \leq \lambda_{min}(\Theta_0) \leq \lambda_{max}(\Theta_0) \leq L$ for some constant $L \geq 1$. Then for all $\Delta$ such that $||\Delta||_F \leq 1/(2L)$, $f(\Delta)$ is well defined and*

$$f(\Delta) \geq \frac{1}{2(L + 1/(2L))^2}||\Delta||_F^2.$$

To simplify the notation, we substitute $\hat{\Sigma}_k, \Sigma_{0k}, \hat{\Theta}_k, \Theta_{0k}$ for $\hat{\Sigma}^{[k]}, \Sigma_0^{[k]}, \hat{\Theta}^{[k]}, \Theta_0^{[k]}$ respectively.

**Proof 1** *Note that $\hat{\Theta}_k$ is the minimum value of the fused graphical Lasso for $k = 1, 2$. Let $\widetilde{\Theta}_k = \alpha_k\hat{\Theta}_k + (1 - \alpha_k)\Theta_{0k}$, and $\alpha_k = \frac{M}{M + ||\hat{\Theta}_k - \Theta_{0k}||_F}$. According to the definitions of $\widetilde{\Theta}_k$, and the convexity of loss function*

$$\begin{aligned} &F_n(\Theta_1, \Theta_2) \\ = \quad &tr(\Theta_1\hat{\Sigma}_1) - \log\det(\Theta_1) + tr(\Theta_2\hat{\Sigma}_2) - \log\det(\Theta_2) \\ &+ \lambda||\Theta_1^-||_1 + \lambda||\Theta_2^-||_1 + \rho||\Theta_1^- - \Theta_2^-||_1, \end{aligned}$$

*we obtain*

$$F_n(\widetilde{\Theta}_1, \widetilde{\Theta}_2) \leq F_n(\Theta_{01}, \Theta_{02}).$$

*That is*

$$\sum_{k=1}^2 \left\{ tr(\widetilde{\Theta}_k - \Theta_{0k})\hat{\Sigma}_k - \left(\log\det(\widetilde{\Theta}_k) - \log\det(\Theta_{0k})\right) + \lambda||\widetilde{\Theta}_k^-||_1 \right\} + \rho||\widetilde{\Theta}_1^- - \widetilde{\Theta}_2^-||_1 \tag{29}$$
$$\leq \quad \lambda||\Theta_{01}^-||_1 + \lambda||\Theta_{02}^-||_1 + \rho||\Theta_{01}^- - \Theta_{02}^-||_1.$$

*Let $\Delta_k = \widetilde{\Theta}_k - \Theta_{0k}$, and*

$$f(\Delta_k) := tr(\Delta_k\Sigma_{0k}) - \left[\log\det(\Delta_k + \Theta_{0k}) - \log\det(\Theta_{0k})\right],$$

*subtracting $tr(\Delta_1(\hat{\Sigma}_1 - \Sigma_{01})) + tr(\Delta_2(\hat{\Sigma}_2 - \Sigma_{02}))$ on the both sides of the inequality (29), we get*

$$f(\Delta_1) + f(\Delta_2) + \lambda||\widetilde{\Theta}_1^-||_1 + \lambda||\widetilde{\Theta}_2^-||_1 + \rho||\widetilde{\Theta}_1^- - \widetilde{\Theta}_2^-||_1 \tag{30}$$
$$\leq \quad -tr(\Delta_1(\hat{\Sigma}_1 - \Sigma_{01})) - tr(\Delta_2(\hat{\Sigma}_2 - \Sigma_{02})) + \lambda||\Theta_{01}^-||_1 + \lambda||\Theta_{02}^-||_1 + \rho||\Theta_{01}^- - \Theta_{02}^-||_1.$$

*For $tr(\Delta_k(\hat{\Sigma}_k - \Sigma_{0k}))$ term, we have*

$$\begin{aligned} |tr(\Delta_k(\hat{\Sigma}_k - \Sigma_{0k}))| \quad &= |G(\Delta_k \circ (\hat{\Sigma}_k - \Sigma_{0k}))| \\ &\leq |G(\Delta_k^- \circ (\hat{\Sigma}_k^- - \Sigma_{0k}^-))| + |G(\Delta_k^+ \circ (\hat{\Sigma}_k^+ - \Sigma_{0k}^+))|, \end{aligned}$$

*where function $G(M)$ takes the summation of all the elements of the matrix $M$, and $\circ$ is Hadamard product. According to Cauchy-Schwarz inequality, on the sets*

$\{\max_k\{||\hat{\Sigma}_k - \Sigma_{0k}||_\infty\} \leq \lambda_0\}$,

$$|G(\Delta_k^- \circ (\hat{\Sigma}_k^- - \Sigma_{0k}^-))| + |G(\Delta_k^+ \circ (\hat{\Sigma}_k^+ - \Sigma_{0k}^+))|$$

$$\leq ||\hat{\Sigma}_k^- - \Sigma_{0k}^-||_\infty ||\Delta_k^-||_1 + ||\hat{\Sigma}_k^+ - \Sigma_{0k}^+||_F ||\Delta_k^+||_F$$

$$\leq \lambda_0 ||\Delta_k^-||_1 + ||\hat{\Sigma}_k^+ - \Sigma_{0k}^+||_F ||\Delta_k^+||_F.$$

*Hence,*

$$\begin{aligned} -tr(\Delta_k(\hat{\Sigma}_k - \Sigma_{0k})) &\leq |tr(\Delta_k(\hat{\Sigma}_k - \Sigma_{0k}))| \\ &\leq \lambda_0 ||\Delta_k^-||_1 + ||\hat{\Sigma}_k^+ - \Sigma_{0k}^+||_F ||\Delta_k^+||_F. \end{aligned} \tag{31}$$

*Next, for $L_k \geq 1$ satisfying condition*

$$1/L_k \leq \lambda_{min}(\Theta_{0k}) \leq \lambda_{max}(\Theta_{0k}) \leq L_k,$$

*we choose $L > 1$ satisfying $1/L \leq 1/L_k$ and $L_k \leq L$, $k = 1, 2$. Based on the definitions of $\Delta_k$ and $\widetilde{\Theta}_k$, we get*

$$||\Delta_k||_F = \alpha_k ||\hat{\Theta}_k - \Theta_{0k}||_F = \frac{||\hat{\Theta}_k - \Theta_{0k}||_F}{M + ||\hat{\Theta}_k - \Theta_{0k}||_F} M, \tag{32}$$

*for arbitrary $M$ in $(0, 1/2L]$. Thus, $||\Delta_k||_F$ is bounded by $M$, i.e., $||\Delta_k||_F \leq M$. For $f(\Delta_k)$ term, based on Lemma 7, we have*

$$f(\Delta_k) \geq c ||\widetilde{\Theta}_k - \Theta_{0k}||_F^2, \tag{33}$$

*where $c = \frac{1}{2(L+1/(2L))^2}$. In particular, we choose $c = 1/(8L^2)$, and the inequality (33) still holds.*

*Using bounds (31) and (33), the inequality (30) turns to be*

$$c||\widetilde{\Theta}_1 - \Theta_{01}||_F^2 + c||\widetilde{\Theta}_2 - \Theta_{02}||_F^2 + \lambda||\widetilde{\Theta}_1^-||_1 + \lambda||\widetilde{\Theta}_2^-||_1 + \rho||\widetilde{\Theta}_1^- - \widetilde{\Theta}_2^-||_1$$

$$\leq \lambda_0||\Delta_1^-||_1 + \lambda_0||\Delta_2^-||_1 + ||\hat{\Sigma}_1^+ - \Sigma_{01}^+||_F ||\Delta_1^+||_F + ||\hat{\Sigma}_2^+ - \Sigma_{02}^+||_F ||\Delta_2^+||_F \tag{34}$$

$$+ \lambda||\Theta_{01}^-||_1 + \lambda||\Theta_{02}^-||_1 + \rho||\Theta_{01}^- - \Theta_{02}^-||_1.$$

*We move some terms of the inequality (34) and combine them to get the following inequality*

$$c||\widetilde{\Theta}_1 - \Theta_{01}||_F^2 + c||\widetilde{\Theta}_2 - \Theta_{02}||_F^2$$

$$+ \lambda\left\{||\widetilde{\Theta}_1^-||_1 - ||\Theta_{01}^-||_1 + ||\widetilde{\Theta}_2^-||_1 - ||\Theta_{02}^-||_1\right\}$$

$$\leq \lambda_0\left\{||\widetilde{\Theta}_1^- - \Theta_{01}^-||_1 + ||\widetilde{\Theta}_2^- - \Theta_{02}^-||_1\right\} + \rho\left\{||\Theta_{01}^- - \Theta_{02}^-||_1 - ||\widetilde{\Theta}_1^- - \widetilde{\Theta}_2^-||_1\right\} \tag{35}$$

$$+ ||\hat{\Sigma}_1^+ - \Sigma_{01}^+||_F ||\widetilde{\Theta}_1^+ - \Theta_{01}^+||_F + ||\hat{\Sigma}_2^+ - \Sigma_{02}^+||_F ||\widetilde{\Theta}_2^+ - \Theta_{02}^+||_F.$$

*Next we need to prove three inequations*:

$$||\widetilde{\Theta}_k^-||_1 - ||\Theta_{0k}^-||_1 \geq ||\Delta_{kS_k^c}^-||_1 - ||\Delta_{kS_k}^-||_1, \tag{36}$$

$$||\widetilde{\Theta}_k^- - \Theta_{0k}^-||_1 \leq ||\Delta_{kS_k^c}^-||_1 + ||\Delta_{kS_k}^-||_1, \tag{37}$$

$$||\Theta_{01}^- - \Theta_{02}^-||_1 - ||\widetilde{\Theta}_1^- - \widetilde{\Theta}_2^-||_1 \leq ||\widetilde{\Theta}_1^- - \Theta_{01}^-||_1 + ||\widetilde{\Theta}_2^- - \Theta_{02}^-||_1. \tag{38}$$

*Because*

$$\begin{aligned}
||\widetilde{\Theta}_k^-||_1 &= ||\Theta_{0k}^- + \Delta_k^-||_1 \\
&= ||\Theta_{0kS_k}^- + \Delta_{kS_k}^-||_1 + ||\Delta_{kS_k^c}^-||_1,
\end{aligned}$$

*and*

$$||\Theta_{0k}^-||_1 = ||\Theta_{0kS_k}^-||_1$$

*hold. Thus,*

$$\begin{aligned}
||\widetilde{\Theta}_k^-||_1 - ||\Theta_{0k}^-||_1 &= ||\Theta_{0kS_k}^- + \Delta_{kS_k}^-||_1 + ||\Delta_{kS_k^c}^-||_1 - ||\Theta_{0kS_k}^-||_1 \\
&\geq ||\Delta_{kS_k^c}^-||_1 - |||\Theta_{0kS_k}^- + \Delta_{kS_k}^-||_1 - ||\Theta_{0kS_k}^-||_1| \\
&\geq ||\Delta_{kS_k^c}^-||_1 - ||\Delta_{kS_k}^-||_1,
\end{aligned}$$

*which proves inequality* (36). *By the triangle inequality, we naturally obtain*

$$\begin{aligned}
||\widetilde{\Theta}_k^- - \Theta_{0k}^-||_1 &= ||\Delta_k^-||_1 \\
&= ||\Delta_{kS_k^c}^- + \Delta_{kS_k}^-||_1 \\
&\leq ||\Delta_{kS_k^c}^-||_1 + ||\Delta_{kS_k}^-||_1.
\end{aligned}$$

*Thus, the inequation* (37) *holds. For inequation* (38), *we have*

$$\begin{aligned}
&||\Theta_{01}^- - \Theta_{02}^-||_1 - ||\widetilde{\Theta}_1^- - \widetilde{\Theta}_2^-||_1 \\
={}& ||\Theta_{01}^- - \widetilde{\Theta}_1^- + \widetilde{\Theta}_1^- - \widetilde{\Theta}_2^- + \widetilde{\Theta}_2^- - \Theta_{02}^-||_1 - ||\widetilde{\Theta}_1^- - \widetilde{\Theta}_2^-||_1 \\
\leq{}& ||\widetilde{\Theta}_1^- - \Theta_{01}^-||_1 + ||\widetilde{\Theta}_2^- - \Theta_{02}^-||_1.
\end{aligned}$$

*Thus, the inequality* (35) *yields*

$$c||\widetilde{\Theta}_1 - \Theta_{01}||_F^2 + c||\widetilde{\Theta}_2 - \Theta_{02}||_F^2$$

$$+\lambda\left\{||\Delta_{1S_1^c}^-||_1 - ||\Delta_{1S_1}^-||_1 + ||\Delta_{2S_2^c}^-||_1 - ||\Delta_{2S_2}^-||_1\right\}$$

$$\leq \quad (\rho + \lambda_0)\left\{||\Delta_{1S_1^c}^-||_1 + ||\Delta_{1S_1}^-||_1 + ||\Delta_{2S_2^c}^-||_1 + ||\Delta_{2S_2}^-||_1\right\}$$

$$+||\hat{\Sigma}_1^+ - \Sigma_{01}^+||_F||\widetilde{\Theta}_1^+ - \Theta_{01}^+||_F + ||\hat{\Sigma}_2^+ - \Sigma_{02}^+||_F||\widetilde{\Theta}_2^+ - \Theta_{02}^+||_F.$$

*By taking* $2(\rho + \lambda_0) < \lambda$, *we conclude that*

$$2c\left\{||\widetilde{\Theta}_1 - \Theta_{01}||_F^2 + ||\widetilde{\Theta}_2 - \Theta_{02}||_F^2\right\} + \lambda\left\{||\Delta_{1S_1^c}^-||_1 + ||\Delta_{2S_2^c}^-||_1\right\}$$

$$\leq \quad 3\lambda\left\{||\Delta_{1S_1}^-||_1 + ||\Delta_{2S_2}^-||_1\right\}$$

$$+2\left\{||\hat{\Sigma}_1^+ - \Sigma_{01}^+||_F||\widetilde{\Theta}_1^+ - \Theta_{01}^+||_F + ||\hat{\Sigma}_2^+ - \Sigma_{02}^+||_F||\widetilde{\Theta}_2^+ - \Theta_{02}^+||_F\right\}.$$

*By the definition of* $\Delta_k$, *we have*

$$\begin{aligned}||\Delta_k^-||_1 = \quad &||\Delta_{kS_k}^- + \Delta_{kS_k^c}^-||_1 \\ \leq \quad &||\Delta_{kS_k}^-||_1 + ||\Delta_{kS_k^c}^-||_1.\end{aligned} \qquad (39)$$

*So we deduce*

$$2c\left\{||\widetilde{\Theta}_1 - \Theta_{01}||_F^2 + ||\widetilde{\Theta}_2 - \Theta_{02}||_F^2\right\} + \lambda\left\{||\Delta_1^-||_1 + ||\Delta_2^-||_1\right\}$$

$$\leq \quad 4\lambda\left\{||\Delta_{1S_1}^-||_1 + ||\Delta_{2S_2}^-||_1\right\}$$

$$+2\left\{||\hat{\Sigma}_1^+ - \Sigma_{01}^+||_F||\widetilde{\Theta}_1^+ - \Theta_{01}^+||_F + ||\hat{\Sigma}_2^+ - \Sigma_{02}^+||_F||\widetilde{\Theta}_2^+ - \Theta_{02}^+||_F\right\}$$

*holds. Since the inequality of arithmetic and geometric means, the inequality* $||\Delta_{kS_k}^-||_1 \leq \sqrt{s_k}||\Delta_{kS_k}^-||_F$ *holds. Thus*

$$2c\left\{||\widetilde{\Theta}_1 - \Theta_{01}||_F^2 + ||\widetilde{\Theta}_2 - \Theta_{02}||_F^2\right\} + \lambda\left\{||\Delta_1^-||_1 + ||\Delta_2^-||_1\right\}$$

$$\leq \quad 4\lambda\left\{\sqrt{s_1}||\Delta_{1S_1}^-||_F + \sqrt{s_2}||\Delta_{2S_2}^-||_F\right\} \qquad (40)$$

$$+2\left\{||\hat{\Sigma}_1^+ - \Sigma_{01}^+||_F||\widetilde{\Theta}_1^+ - \Theta_{01}^+||_F + ||\hat{\Sigma}_2^+ - \Sigma_{02}^+||_F||\widetilde{\Theta}_2^+ - \Theta_{02}^+||_F\right\}.$$

*Using $xy \leq (x^2 + y^2)/2$, the inequality* (40) *infer that*

$$2c\Big\{||\widetilde{\Theta}_1 - \Theta_{01}||_F^2 + ||\widetilde{\Theta}_2 - \Theta_{02}||_F^2\Big\} + \lambda\Big\{||\Delta_1^-||_1 + ||\Delta_2^-||_1\Big\}$$

$$\leq \quad \frac{1}{2}\left(c||\Delta_{1S_1}^-||_F^2 + \frac{16\lambda^2 s_1}{c} + c||\Delta_{2S_2}^-||_F^2 + \frac{16\lambda^2 s_2}{c}\right)$$

$$+ \frac{1}{2}\left(c||\widetilde{\Theta}_1^+ - \Theta_{01}^+||_F^2 + \frac{4||\hat{\Sigma}_1^+ - \Sigma_{01}^+||_F^2}{c} + c||\widetilde{\Theta}_2^+ - \Theta_{02}^+||_F^2 + \frac{4||\hat{\Sigma}_2^+ - \Sigma_{02}^+||_F^2}{c}\right).$$

*Because*

$$c||\widetilde{\Theta}_k^+ - \Theta_{0k}^+||_F^2 + c||\Delta_{kS_k}^-||_F^2 \leq \quad \Big\{c||\widetilde{\Theta}_k^+ - \Theta_{0k}^+||_F^2 + c||\Delta_k^-||_F^2\Big\}$$

$$+ \Big\{c||\Delta_{kS_k}^-||_F^2 + c||\Delta_{kS_k^c}^-||_F^2 + c||\Delta_k^+||_F^2\Big\} \tag{41}$$

$$= \quad 2c||\Delta_k||_F^2,$$

*we obtain*

$$2c\Big\{||\widetilde{\Theta}_1 - \Theta_{01}||_F^2 + ||\widetilde{\Theta}_2 - \Theta_{02}||_F^2\Big\} + \lambda\Big\{||\Delta_1^-||_1 + ||\Delta_2^-||_1\Big\}$$

$$\leq \quad c\Big\{||\Delta_1||_F^2 + ||\Delta_2||_F^2\Big\} + \frac{8\lambda^2(s_1 + s_2)}{c} + \frac{2||\hat{\Sigma}_1^+ - \Sigma_{01}^+||_F^2}{c} + \frac{2||\hat{\Sigma}_2^+ - \Sigma_{02}^+||_F^2}{c}.$$

*Thus,*

$$c\Big\{||\Delta_1||_F^2 + ||\Delta_2||_F^2\Big\} + \lambda\Big\{||\Delta_1^-||_1 + ||\Delta_2^-||_1\Big\}$$

$$\leq \quad \frac{8\lambda^2(s_1 + s_2)}{c} + \frac{2||\hat{\Sigma}_1^+ - \Sigma_{01}^+||_F^2}{c} + \frac{2||\hat{\Sigma}_2^+ - \Sigma_{02}^+||_F^2}{c}. \tag{42}$$

*Based on the inequality $||\hat{\Sigma}_k^+ - \Sigma_{0k}^+||_F \leq \sqrt{p}||\hat{\Sigma}_k^+ - \Sigma_{0k}^+||_\infty$, we have*

$$c\{||\Delta_1||_F^2 + ||\Delta_2||_F^2\} + \lambda\{||\Delta_1^-||_1 + ||\Delta_2^-||_1\} \leq \frac{8\lambda^2(s_1 + s_2)}{c} + \frac{4p\lambda_0^2}{c}. \tag{43}$$

*Next, we prove that substituting $\hat{\Theta}_k$ for $\widetilde{\Theta}_k$, the conclusion still holds. According to the condition,*

$$||\Delta_1||_F^2 + ||\Delta_2||_F^2 \leq \quad \frac{\lambda_0}{2cL} \leq \frac{\lambda}{4cL} \leq \frac{1}{32L^2}.$$

*Taking $M = 1/(2\sqrt{2}L) < 1/2L$, we have*

$$||\Delta_1||_F^2 + ||\Delta_2||_F^2 \leq \quad M^2/4.$$

*Thus, $||\Delta_k||_F$ is bounded by $M/2$. In addition,*

$$||\hat{\Theta}_k - \Theta_{0k}||_F = \frac{M||\Delta_k||_F}{M - ||\Delta_k||_F},$$

*which means $||\hat{\Theta}_k - \Theta_{0k}||_F$ is monotone increasing function of $||\Delta_k||_F$ on set $(0, M)$. We obtain*

that $||\hat{\Theta}_k - \Theta_{0k}||_F \leq M$. Therefore, we can substitute $\hat{\Theta}_k$ for $\tilde{\Theta}_k$, and that leads to the inequality (43) holds for $\hat{\Theta}_k$.

According to inequality (43), we get

$$
||\hat{\Theta}_k - \Theta_{0k}||_F^2 \leq \quad \frac{8\lambda^2(s_1 + s_2)}{c^2} + \frac{4p\lambda_0^2}{c^2}
$$

$$
\leq \quad \frac{\lambda^2(8s_1 + 8s_2 + p)}{c^2},
$$

and

$$
||\hat{\Theta}_k^- - \Theta_{0k}^-||_1 \leq \quad \frac{8\lambda(s_1 + s_2)}{c} + \frac{4p\lambda_0^2}{\lambda c}
$$

$$
\leq \quad \frac{\lambda(8s_1 + 8s_2 + p)}{c}.
$$

Thus, we conclude the upper bound of $\sum_{k=1}^{2} |||\hat{\Theta}_k - \Theta_{0k}|||_1$,

$$
\sum_{k=1}^{2} |||\hat{\Theta}_k - \Theta_{0k}|||_1 \leq \quad \sum_{k=1}^{2} \left( ||\hat{\Theta}_k^+ - \Theta_{0k}^+||_\infty + ||\hat{\Theta}_k^- - \Theta_{0k}^-||_1 \right)
$$

$$
\leq \quad \sum_{k=1}^{2} \left( ||\hat{\Theta}_k - \Theta_{0k}||_F + ||\hat{\Theta}_k^- - \Theta_{0k}^-||_1 \right)
$$

$$
\leq \quad \frac{2\lambda\sqrt{8s_1 + 8s_2 + p}}{c} + \frac{2\lambda(8s_1 + 8s_2 + p)}{c}
$$

$$
\leq \quad \frac{4\lambda(8s_1 + 8s_2 + p)}{c}.
$$

## Proof of Theorem 2

**Proof 2** *The minimizer* $(\hat{\Theta}_R^{[1]}, \hat{\Theta}_R^{[2]})$ *satisfying inequality* (42), *that is*

$$
c\left\{ ||\hat{\Theta}_R^{[1]} - \Theta_{R0}^{[1]}||_F^2 + ||\hat{\Theta}_R^{[2]} - \Theta_{R0}^{[2]}||_F^2 \right\} + \lambda \left\{ ||(\hat{\Theta}_R^{[1]} - \Theta_{R0}^{[1]})^-||_1 + ||(\hat{\Theta}_R^{[2]} - \Theta_{R0}^{[2]})^-||_1 \right\}
$$

$$
\leq \quad \frac{8\lambda^2(s_1 + s_2)}{c} + \frac{2||(\hat{R}^{[1]} - R_0^{[1]})^+||_F^2}{c} + \frac{2||(\hat{R}^{[2]} - R_0^{[2]})^+||_F^2}{c}.
$$

*The diagonal elements of* $\hat{R}^{[k]}$ *and* $R_0^{[k]}$ *are all* 1. *Thus*

$$
c\left\{ ||\hat{\Theta}_R^{[1]} - \Theta_{R0}^{[1]}||_F^2 + ||\hat{\Theta}_R^{[2]} - \Theta_{R0}^{[2]}||_F^2 \right\} + \lambda \left\{ ||(\hat{\Theta}_R^{[1]} - \Theta_{R0}^{[1]})^-||_1 + ||(\hat{\Theta}_R^{[2]} - \Theta_{R0}^{[2]})^-||_1 \right\}
$$

$$
\leq \quad \frac{8\lambda^2(s_1 + s_2)}{c}.
$$

*Moreover, for the conclusion of the $l_1$-operator norm, we get*

$$
|||\hat{\Theta}_R^{[1]} - \Theta_{R0}^{[1]}|||_1 + |||\hat{\Theta}_R^{[2]} - \Theta_{R0}^{[2]}|||_1
$$

$$
\leq \quad \sum_{k=1}^{2} \left( ||(\hat{\Theta}_R^{[k]} - \Theta_{R0}^{[k]})^+||_\infty + ||(\hat{\Theta}_R^{[k]} - \Theta_{R0}^{[k]})^-||_1 \right)
$$

$$
\leq \quad \sum_{k=1}^{2} \left( ||\hat{\Theta}_R^{[k]} - \Theta_{R0}^{[k]}||_F + ||(\hat{\Theta}_R^{[k]} - \Theta_{R0}^{[k]})^-||_1 \right)
$$

$$
\leq \quad \frac{32\lambda(s_1 + s_2)}{c}.
$$

*For the minimizer $(\hat{\Theta}_w^{[1]}, \hat{\Theta}_w^{[2]})$, following inequality holds*

$$
\begin{aligned}
&|||\hat{\Theta}_R^{[k]} - \Theta_{R0}^{[k]}|||_1 \\
= \quad &|||\hat{W}^{[k]}\hat{\Theta}_w^{[k]}\hat{W}^{[k]} - W_0^{[k]}\Theta_{w0}^{[k]}W_0^{[k]}|||_1 \\
\leq \quad &||\hat{W}^{[k]}||_\infty^2 |||\hat{\Theta}_w^{[k]} - \Theta_{w0}^{[k]}|||_1 + ||\hat{W}^{[k]} - W_0^{[k]}||_\infty |||\Theta_{w0}^{[k]}|||_1 ||\hat{W}^{[k]}||_\infty \\
&+ ||W_0^{[k]}||_\infty |||\Theta_{w0}^{[k]}|||_1 ||\hat{W}^{[k]} - W_0^{[k]}||_\infty.
\end{aligned}
\tag{44}
$$

*To draw the conclusion, we have the following facts:*

- *The Sub-Gaussian vector with covariance $\Sigma_0^{[k]}$ implies that $\sqrt{n/\log p}||(\hat{\Sigma}^{[k]} - \Sigma_0^{[k]})||_\infty$ is bounded in probability.*

- *The eigenvalues of $\Theta_{w0}^{[k]}$ are bounded by a constant.*

  *Thus, $|||\hat{\Theta}_R^{[k]} - \Theta_{R0}^{[k]}|||_1$ and $|||\hat{\Theta}_w^{[k]} - \Theta_{w0}^{[k]}|||_1$ share the same boundary.*

## Proof of Theorem 3

**Proof 3** *Similarly, $\hat{\Theta}_k$ are the minimum value of the fused graphical Lasso for $k = 1, 2, \cdots, K$. Let $\tilde{\Theta}_k = \alpha_k \hat{\Theta}_k + (1 - \alpha_k)\Theta_{0k}$, and $\alpha_k = \frac{M}{M + ||\hat{\Theta}_k - \Theta_{0k}||_F}$. Denotes*

$$
F_n(\Theta_1, \cdots, \Theta_K) = \sum_{k=1}^{K} \left\{ tr(\Theta_k \hat{\Sigma}_k) - \log \det(\Theta_k) \right\} + \lambda \sum_{k=1}^{K} ||\Theta_k^-||_1 + \rho \sum_{k<k'} ||\Theta_k^- - \Theta_{k'}^-||_1,
$$

*we obtain*

$$
F_n(\tilde{\Theta}_1, \tilde{\Theta}_2, \cdots, \tilde{\Theta}_K) \leq F_n(\Theta_{01}, \Theta_{02}, \cdots, \Theta_{0K}).
$$

*Thus,*

$$\sum_{k=1}^{K} \left\{ tr(\widetilde{\Theta}_k - \Theta_{0k})\hat{\Sigma}_k - \left(\log \det(\widetilde{\Theta}_k) - \log \det(\Theta_{0k})\right) + \lambda ||\widetilde{\Theta}_k^-||_1 \right\}$$

$$+ \rho \sum_{k<k'} ||\widetilde{\Theta}_k^- - \widetilde{\Theta}_{k'}^-||_1$$

$$\leq \quad \lambda \sum_{k=1}^{K} ||\Theta_{0k}^-||_1 + \rho \sum_{k<k'} ||\Theta_{0k}^- - \Theta_{0k'}^-||_1.$$

*Using the notations that* $\Delta_k = \widetilde{\Theta}_k - \Theta_{0k}$ *and*

$$f(\Delta_k) := tr(\Delta_k \Sigma_{0k}) - \left[\log \det(\Delta_k + \Theta_{0k}) - \log \det(\Theta_{0k})\right],$$

*we yield the following expression*

$$\sum_{k=1}^{K} f(\Delta_k) + \lambda \sum_{k=1}^{K} ||\widetilde{\Theta}_k^-||_1 + \rho \sum_{k<k'} ||\widetilde{\Theta}_k^- - \widetilde{\Theta}_{k'}^-||_1$$

$$\leq \quad -\sum_{k=1}^{K} \left( tr(\Delta_k(\hat{\Sigma}_k - \Sigma_{0k})) \right) - tr(\Delta_2(\hat{\Sigma}_2 - \Sigma_{02})) \quad\quad (45)$$

$$+ \lambda \sum_{k=1}^{K} ||\Theta_{0k}^-||_1 + \rho \sum_{k<k'} ||\Theta_{0k}^- - \Theta_{0k'}^-||_1.$$

*For* $L_k \geq 1$, $k = 1, 2, \cdots, K$, *the minimum and maximum eigenvalues of* $\Theta_{0k}$ *hold that*

$$1/L_k \leq \lambda_{min}(\Theta_{0k}) \leq \lambda_{max}(\Theta_{0k}) \leq L_k.$$

*For multiple case, we select a constant* $L$ *satisfying* $1/L \leq 1/L_k$ *and* $L_k \leq L$. *By similar analysis, for* $M$ *in* $(0, 1/2L]$, *the inequality* (32) *and the inequality* (33) *still hold.*

*For K groups data, based on the inequalities* (31) *and* (33). *Then, the inequality* (45) *turns to be*

$$c \sum_{k=1}^{K} ||\widetilde{\Theta}_k - \Theta_{0k}||_F^2 + \lambda \sum_{k=1}^{K} ||\widetilde{\Theta}_k^-||_1 + \rho \sum_{k<k'} ||\widetilde{\Theta}_k^- - \widetilde{\Theta}_{k'}^-||_1$$

$$\leq \quad \sum_{k=1}^{K} \left\{ \lambda_0 ||\Delta_k^-||_1 + ||\hat{\Sigma}_k^+ - \Sigma_{0k}^+||_F ||\Delta_k^+||_F \right\} + \lambda \sum_{k=1}^{K} ||\Theta_{0k}^-||_1 + \rho \sum_{k<k'} ||\Theta_{0k}^- - \Theta_{0k'}^-||_1.$$

*Thus,*

$$c \sum_{k=1}^{K} ||\widetilde{\Theta}_k - \Theta_{0k}||_F^2 + \lambda \sum_{k=1}^{K} \left\{ ||\widetilde{\Theta}_k^-||_1 - ||\Theta_{0k}^-||_1 \right\}$$

$$\leq \quad \lambda_0 \sum_{k=1}^{K} ||\widetilde{\Theta}_k^- - \Theta_{0k}^-||_1 + \rho \sum_{k<k'} \left\{ ||\Theta_{0k}^- - \Theta_{0k'}^-||_1 - ||\widetilde{\Theta}_k^- - \widetilde{\Theta}_{k'}^-||_1 \right\} \quad\quad (46)$$

$$+ \sum_{k=1}^{K} \left\{ ||\hat{\Sigma}_k^+ - \Sigma_{0k}^+||_F ||\widetilde{\Theta}_k^+ - \Theta_{0k}^+||_F \right\}.$$

*When k = 1, 2, · · ·, K, the inequations* (36) *and* (37) *still hold. Similarly, we have the following inequality*

$$
\begin{aligned}
&||\Theta_{0k}^- - \Theta_{0k'}^-||_1 - ||\widetilde{\Theta}_k^- - \widetilde{\Theta}_{k'}^-||_1 \\
=\ & ||\Theta_{0k}^- - \widetilde{\Theta}_k^- + \widetilde{\Theta}_k^- - \widetilde{\Theta}_{k'}^- + \widetilde{\Theta}_{k'}^- - \Theta_{0k'}^-||_1 - ||\widetilde{\Theta}_k^- - \widetilde{\Theta}_{k'}^-||_1 \\
\leq\ & ||\widetilde{\Theta}_k^- - \Theta_{0k}^-||_1 + ||\widetilde{\Theta}_{k'}^- - \Theta_{0k'}^-||_1.
\end{aligned}
\tag{47}
$$

*Thus, by the* Eqs (36), (37) *and* (47) *the inequality* (46) *yields*

$$
\begin{aligned}
& c\sum_{k=1}^K ||\widetilde{\Theta}_k - \Theta_{0k}||_F^2 + \lambda\sum_{k=1}^K \Big\{ ||\Delta_{kS_k^c}^-||_1 - ||\Delta_{kS_k}^-||_1 \Big\} \\
\leq\ & \lambda_0 \sum_{k=1}^K \Big\{ ||\Delta_{kS_k^c}^-||_1 + ||\Delta_{kS_k}^-||_1 \Big\} \\
& + \rho \sum_{k<k'} \Big\{ ||\Delta_{kS_k^c}^-||_1 + ||\Delta_{kS_k}^-||_1 + ||\Delta_{k'S_{k'}^c}^-||_1 + ||\Delta_{k'S_{k'}}^-||_1 \Big\} \\
& + \sum_{k=1}^K \Big\{ ||\hat{\Sigma}_k^+ - \Sigma_{0k}^+||_F ||\widetilde{\Theta}_k^+ - \Theta_{0k}^+||_F \Big\} \\
\leq\ & \left( \frac{K(K-1)}{2}\rho + \lambda_0 \right) \sum_{k=1}^K \Big\{ ||\Delta_{kS_k^c}^-||_1 + ||\Delta_{kS_k}^-||_1 \Big\} \\
& + \sum_{k=1}^K \Big\{ ||\hat{\Sigma}_k^+ - \Sigma_{0k}^+||_F ||\widetilde{\Theta}_k^+ - \Theta_{0k}^+||_F \Big\}.
\end{aligned}
$$

*Since K is a fixed constant, and* $2\left(\frac{K(K-1)}{2}\rho + \lambda_0\right) < \lambda$, *we can obtain*

$$
\begin{aligned}
& 2c\sum_{k=1}^K ||\widetilde{\Theta}_k - \Theta_{0k}||_F^2 + \lambda\sum_{k=1}^K ||\Delta_{kS_k^c}^-||_1 \\
\leq\ & 3\lambda\sum_{k=1}^K ||\Delta_{kS_k}^-||_1 + 2\sum_{k=1}^K \Big\{ ||\hat{\Sigma}_k^+ - \Sigma_{0k}^+||_F ||\widetilde{\Theta}_k^+ - \Theta_{0k}^+||_F \Big\}.
\end{aligned}
$$

*On the basis of the inequality* (39), *we deduce*

$$
\begin{aligned}
& 2c\sum_{k=1}^K ||\widetilde{\Theta}_k - \Theta_{0k}||_F^2 + \lambda\sum_{k=1}^K ||\Delta_k^-||_1 \\
\leq\ & 4\lambda\sum_{k=1}^K ||\Delta_{kS_k}^-||_1 + 2\sum_{k=1}^K \Big\{ ||\hat{\Sigma}_k^+ - \Sigma_{0k}^+||_F ||\widetilde{\Theta}_k^+ - \Theta_{0k}^+||_F \Big\}
\end{aligned}
$$

*holds. In addition, one can get the inequality $||\Delta_{kS_k}^-||_1 \leq \sqrt{s_k}||\Delta_{kS_k}^-||_F$. Thus*

$$
2c\sum_{k=1}^{K}||\widetilde{\Theta}_k - \Theta_{0k}||_F^2 + \lambda\sum_{k=1}^{K}||\Delta_k^-||_1
$$

$$
\leq \quad 4\lambda\sum_{k=1}^{K}(\sqrt{s_k}||\Delta_{kS_k}^-||_F) + 2\sum_{k=1}^{K}\left\{||\hat{\Sigma}_k^+ - \Sigma_{0k}^+||_F||\widetilde{\Theta}_k^+ - \Theta_{0k}^+||_F\right\}.
$$

(48)

*Based on $xy \leq (x^2 + y^2)/2$ and the inequality (41), the inequality (48) infer that*

$$
2c\sum_{k=1}^{K}||\widetilde{\Theta}_k - \Theta_{0k}||_F^2 + \lambda\sum_{k=1}^{K}||\Delta_k^-||_1
$$

$$
\leq \quad \frac{1}{2}\sum_{k=1}^{K}\left(c||\Delta_{kS_k}^-||_F^2 + \frac{16\lambda^2 s_k}{c}\right) + \frac{1}{2}\sum_{k=1}^{K}\left(c||\widetilde{\Theta}_k^+ - \Theta_{0k}^+||_F^2 + \frac{4||\hat{\Sigma}_k^+ - \Sigma_{0k}^+||_F^2}{c}\right)
$$

$$
\leq \quad c\sum_{k=1}^{K}||\Delta_k||_F^2 + \frac{8\lambda^2\sum_{k=1}^{K}s_k}{c} + \frac{2\sum_{k=1}^{K}||\hat{\Sigma}_k^+ - \Sigma_{0k}^+||_F^2}{c}.
$$

*Thus,*

$$
c\sum_{k=1}^{K}||\Delta_k||_F^2 + \lambda\sum_{k=1}^{K}||\Delta_k^-||_1 \leq \frac{8\lambda^2\sum_{k=1}^{K}s_k}{c} + \frac{2\sum_{k=1}^{K}||\hat{\Sigma}_k^+ - \Sigma_{0k}^+||_F^2}{c}.
$$

(49)

*Using the relation between the Frobenius norm and the supremum norm, we have*

$$
c\sum_{k=1}^{K}||\Delta_k||_F^2 + \lambda\sum_{k=1}^{K}||\Delta_k^-||_1 \leq \frac{8\lambda^2\sum_{k=1}^{K}s_k}{c} + \frac{2Kp\lambda_0^2}{c}.
$$

(50)

*According to the inequality (50), we get*

$$
\sum_{k=1}^{K}||\Delta_k||_F^2 \leq \frac{\lambda_0}{2cL}.
$$

*According to $\lambda_0 \leq \lambda/2$ and the condition $\lambda \leq c/8L$, we get*

$$
\sum_{k=1}^{K}||\Delta_k||_F^2 \leq \frac{1}{32L^2}.
$$

*Taking $M = 1/(2\sqrt{2}L) < 1/2L$, we have*

$$
\sum_{k=1}^{K}||\Delta_k||_F^2 \leq M^2/4.
$$

*Thus, $||\Delta_k||_F$ is bounded by $M/2$. Further, we can derive $||\hat{\Theta}_k - \Theta_{0k}||_F \leq M$ which means that we can substitute $\hat{\Theta}_k$ for $\widetilde{\Theta}_k$, and that leads to the inequality (50) holds for $\hat{\Theta}_k$, i.e.*

$$
c\sum_{k=1}^{K}||\hat{\Theta}_k - \Theta_{0k}||_F^2 + \lambda\sum_{k=1}^{K}||(\hat{\Theta}_k - \Theta_{0k})^-||_1 \leq \frac{8\lambda^2\sum_{k=1}^{K}s_k}{c} + \frac{2Kp\lambda_0^2}{c},
$$

*That implies*

$$
\begin{aligned}
\sum_{k=1}^{K} |||\hat{\Theta}_k - \Theta_{0k}|||_1 &\leq \sum_{k=1}^{K} \left( ||\hat{\Theta}_k^+ - \Theta_{0k}^+||_\infty + ||\hat{\Theta}_k^- - \Theta_{0k}^-||_1 \right) \\
&\leq \sum_{k=1}^{K} \left( ||\hat{\Theta}_k - \Theta_{0k}||_F + ||\hat{\Theta}_k^- - \Theta_{0k}^-||_1 \right) \\
&\leq K \left[ \frac{\lambda \sqrt{8 \sum_{k=1}^{K} s_k + \frac{Kp}{2}}}{c} + \frac{\lambda \left( 8 \sum_{k=1}^{K} s_k + \frac{Kp}{2} \right)}{c} \right] \\
&\leq \frac{2K\lambda \left( 8 \sum_{k=1}^{K} s_k + \frac{Kp}{2} \right)}{c},
\end{aligned}
$$

*which completes the proof.*

## Proof of Theorem 4

**Proof 4** *We get from* (49)

$$
\begin{aligned}
c \sum_{k=1}^{K} ||\hat{\Theta}_R^{[k]} - \Theta_{R0}^{[k]}||_F^2 + \lambda \sum_{k=1}^{K} ||(\hat{\Theta}_R^{[k]} - \Theta_{R0}^{[k]})^-||_1 \\
\leq \frac{8\lambda^2 \sum_{k=1}^{K} s_k}{c} + \frac{2 \sum_{k=1}^{K} ||(\hat{\Theta}_R^{[k]} - \Theta_{R0}^{[k]})^+||_F^2}{c},
\end{aligned}
$$

*and similarly derive*

$$
c \sum_{k=1}^{K} ||\hat{\Theta}_R^{[k]} - \Theta_{R0}^{[k]}||_F^2 + \lambda \sum_{k=1}^{K} ||(\hat{\Theta}_R^{[k]} - \Theta_{R0}^{[k]})^-||_1 \leq \frac{8\lambda^2 \sum_{k=1}^{K} s_k}{c}.
$$

*Using*

$$
\sum_{k=1}^{K} |||\hat{\Theta}_R^{[k]} - \Theta_{R0}^{[k]}|||_1 \leq \sum_{k=1}^{K} \left( ||\hat{\Theta}_R^{[k]} - \Theta_{R0}^{[k]}||_F + ||(\hat{\Theta}_R^{[k]} - \Theta_{R0}^{[k]})^-||_1 \right)
$$

*we have*

$$
\sum_{k=1}^{K} |||\hat{\Theta}_R^{[k]} - \Theta_{R0}^{[k]}|||_1 \leq \frac{16K\lambda \sum_{k=1}^{K} s_k}{c}.
$$

*At last, using the inequality* (44), *based on the analysis of the upper bound of* $||W_0^{[k]}||_\infty$ *and* $||\hat{W}^{[k]}||_\infty$, *and the convergence rate of* $||(\hat{\Sigma}^{[k]} - \Sigma_0^{[k]})||_\infty$, *we draw the conclusion that*

$$
\sum_{k=1}^{K} |||\hat{\Theta}_w^{[k]} - \Theta_0^{[k]}|||_1 \frac{16K\lambda \sum_{k=1}^{K} s_k}{c}.
$$

## Proof of Theorem 5

**Proof 5** *First of all, we prove that the remainder converge in probability with a $1/\sqrt{n}$ convergence rate. On account of Theorem 1, we get*

$$||rem||_\infty \leq \sum_{k=1}^{2}||(\hat{\Theta}^{[k]} - \Theta_0^{[k]})(\hat{\Sigma}^{[k]} - \Sigma_0^{[k]})\Theta_0^{[k]}||_\infty + \sum_{k=1}^{2}||(\hat{\Theta}^{[k]} - \Theta_0^{[k]})(\hat{\Sigma}^{[k]}\hat{\Theta}^{[k]} - \mathbf{I}_p)||_\infty$$

*Define*

$$l(\Theta) = \sum_{k=1}^{2}\{tr(\hat{\Sigma}^{[k]}\Theta^{[k]}) - \log\det(\Theta^{[k]})\} + \lambda\sum_{k=1}^{2}||(\Theta^{[k]})^-||_1 + \rho||(\Theta^{[1]} - \Theta^{[2]})^-||_1.$$

*By the Karush-Kuhn-Tucker (KKT) conditions, we yield*

$$\hat{\Sigma}^{[1]} - (\hat{\Theta}^{[1]})^{-1} + (\lambda + \rho)\hat{Z}^{[1]} = 0, \tag{51}$$

*and*

$$\hat{\Sigma}^{[2]} - (\hat{\Theta}^{[2]})^{-1} + (\lambda - \rho)\hat{Z}^{[2]} = 0, \tag{52}$$

*where $\hat{Z}_{ij}^{[k]} = sign(\hat{\Theta}_{ij}^{[k]})$ if $\hat{\Theta}_{ij}^{[k]} \neq 0$, and satisfying $||\hat{Z}^{[k]}||_\infty \leq 1$. Multiplying by $\hat{\Theta}^{[1]}$ on the* Eq *(51), we get*

$$\mathbf{I}_p - \hat{\Sigma}^{[1]}\hat{\Theta}^{[1]} = (\lambda + \rho)\hat{Z}^{[1]}\hat{\Theta}^{[1]}.$$

*Similarly, we have*

$$\mathbf{I}_p - \hat{\Sigma}^{[2]}\hat{\Theta}^{[2]} = (\lambda - \rho)\hat{Z}^{[2]}\hat{\Theta}^{[2]}.$$

*Thus,*

$$||rem||_\infty \leq \quad \sum_{k=1}^{2}|||(\hat{\Theta}^{[k]} - \Theta_0^{[k]})|||_1||(\hat{\Sigma}^{[k]} - \Sigma_0^{[k]})||_\infty|||\Theta_0^{[k]}|||_1$$

$$+ (\lambda + \rho)\sum_{k=1}^{2}|||(\hat{\Theta}^{[k]} - \Theta_0^{[k]})|||_1||\hat{Z}^{[k]}||_\infty|||\hat{\Theta}^{[k]}|||_1.$$

*To draw the conclusion, we have*

$$|||(\hat{\Theta}^{[k]} - \Theta_0^{[k]})|||_1 \leq b(p + s)\lambda, \tag{53}$$

*where b is a constant and is related to L. According to the Schwarz inequality and Weyl inequality, we get*

$$|||\Theta_0^{[k]}|||_1 \leq \sqrt{d+1}\Lambda_{\max}(\Theta_0^{[k]}). \tag{54}$$

*The bound of $|||\hat{\Theta}^{[k]}|||_1$ is derived by*

$$|||\hat{\Theta}^{[k]}|||_1 \leq |||\hat{\Theta}^{[k]} - \Theta_0^{[k]}|||_1 + |||\Theta_0^{[k]}|||_1. \tag{55}$$

*According to the rate of $\lambda$, we conclude that*

$$|||\hat{\Theta}^{[k]}|||_1 \leq \sqrt{d+1}\Lambda_{\max}(\Theta_0^{[k]}). \tag{56}$$

*Besides, the Sub-Gaussian random vector with covariance $\Sigma_0^{[k]}$ implies that*

$||\hat{\Sigma}^{[k]} - \Sigma_0^{[k]}||_\infty = O_p(\sqrt{\log(p)/n})$, where $O_P$ denotes bounded in probability. We get

$$
\begin{aligned}
||rem||_\infty \quad \le \quad & \frac{4\lambda(8s_1 + 8s_2 + p)}{c}\sqrt{\frac{\log p}{n}}\sqrt{d+1}\max\{\Lambda_{\max}(\Theta_0^{[1]}), \Lambda_{\max}(\Theta_0^{[2]})\} \\
& + (\lambda + \rho)\frac{4\lambda(8s_1 + 8s_2 + p)}{c}\sqrt{d+1}\max\{\Lambda_{\max}(\Theta_0^{[1]}), \Lambda_{\max}(\Theta_0^{[2]})\}.
\end{aligned}
$$

*For $\lambda \asymp \rho$, $||rem||_\infty$ is bounded by $\tilde{b}(p+s)\sqrt{d+1}\lambda^2$ in probability, where $\tilde{b}$ is a constant related to L. Based on the condition $(p+s)\sqrt{d} = o(\sqrt{n}/\log p)$, $||rem||_\infty = o_p(1/\sqrt{n})$. According to the bounded fourth moments of $(\hat{\Theta}^{[k]})_{ii}(\hat{\Theta}^{[k]})_{jj} + (\hat{\Theta}^{[k]})_{ij}^2$ and Lindeberg central limit theorem, we complete the proof of the Theorem 5.*

## Proof of Theorem 6

**Proof 6** *The conclusions of Theorem 6 can be obtained from the arguments (53)–(56). For weighted version, $||rem||_\infty$ can be bounded by $\tilde{b}s\sqrt{d+1}\lambda^2$, which completes the proof.*

## Author Contributions

**Data curation:** Hu Yang.

**Formal analysis:** Hu Yang.

**Funding acquisition:** Qiuyan Zhang.

**Software:** Lingrui Li.

**Writing – original draft:** Qiuyan Zhang.

**Writing – review & editing:** Qiuyan Zhang.

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
