## [Decision Letter · Decision Letter 0]

3 Apr 2024

PONE-D-23-38136Application of fused graphical lasso to statistical inference for multiple sparse precision matricesPLOS ONE

Dear Dr. Zhang,

Thank you for submitting your manuscript to PLOS ONE. After careful consideration, we feel that it has merit but does not fully meet PLOS ONE’s publication criteria as it currently stands. Therefore, we invite you to submit a revised version of the manuscript that addresses the points raised during the review process.

We look forward to receiving your revised manuscript.

Kind regards,

Debo Cheng

Academic Editor

PLOS ONE

Journal Requirements:

Did you know that depositing data in a repository is associated with up to a 25% citation advantage (https://doi.org/10.1371/journal.pone.0230416)? If you’ve not already done so, consider depositing your raw data in a repository to ensure your work is read, appreciated and cited by the largest possible audience. You’ll also earn an Accessible Data icon on your published paper if you deposit your data in any participating repository (https://plos.org/open-science/open-data/#accessible-data).

3. Please note that PLOS ONE has specific guidelines on code sharing for submissions in which author-generated code underpins the findings in the manuscript. In these cases, all author-generated code must be made available without restrictions upon publication of the work. 

Please review our guidelines at https://journals.plos.org/plosone/s/materials-and-software-sharing#loc-sharing-code and ensure that your code is shared in a way that follows best practice and facilitates reproducibility and reuse.

4. Please update your submission to use the PLOS LaTeX template. The template and more information on our requirements for LaTeX submissions can be found at http://journals.plos.org/plosone/s/latex.

"Q.Y. Zhang was supported by NSFC 12201430."

7. Thank you for uploading your study's underlying data set. Unfortunately, the repository you have noted in your Data Availability statement does not qualify as an acceptable data repository according to PLOS's standards.

8. Your abstract cannot contain citations. Please only include citations in the body text of the manuscript, and ensure that they remain in ascending numerical order on first mention.

**Additional Editor Comments:**

Both reviewers acknowledged the paper's contributions, yet they also highlighted certain drawbacks. Therefore, they recommend a major revision to address these issues.

Reviewers' comments:

Reviewer's Responses to Questions

**Comments to the Author**

1. Is the manuscript technically sound, and do the data support the conclusions?

Reviewer #1: Yes

Reviewer #2: Yes

2. Has the statistical analysis been performed appropriately and rigorously? 

Reviewer #1: Yes

Reviewer #2: Yes

3. Have the authors made all data underlying the findings in their manuscript fully available?

Reviewer #1: Yes

Reviewer #2: Yes

4. Is the manuscript presented in an intelligible fashion and written in standard English?

Reviewer #1: Yes

Reviewer #2: Yes

5. Review Comments to the Author

Reviewer #1: The authors study the problem of estimating multiple precision matrices from

multiple populations. They propose to use the fused graphical lasso (FGL) method, which is a lasso penalty on sparsity of precision matrices plus

a refined lasso penalty

on the two precision matrices that restrains the

similar structure across multiple groups.

They obtain some inequalities for multiple estimators of FGL models in terms of $L_1$ and Frobenius matrix norms

under several conditions.

Built on the work of Jankov´a and van de Geer (2015) who investigated a de-biasing

technology to obtain a new consistent estimator with known distribution, the authors extend the statistical inference problem to multiple populations, and propose the de-biasing

FGL estimators. The corresponding asymptotic property of de-biasing FGL estimators is provided. A

simulation study and an application of the diffuse large B-cell lymphoma data demonstrate theoretical results.

It is a nice contribution. Some comments are in the attached file.

Reviewer #2: The authors use the fused graphical lasso (FGL) method to estimate multiple precision matrices from multiple populations simultaneously. In high-dimensional setting, an oracle inequality is provided for FGL estimators, which is necessary to establish the central limit law. They not only focus on point estimation of precision matrix, but also work on hypothesis testing for a linear combination of the entries of multiple precision matrices. They extend the Jankov$\\acute{a}$ and van de Geer's [Confidence intervals for high-dimensional inverse covariance estimation, {\\it Electron. J. Stat.} {\\bf 9}(1) (2015) 1205-1229.] de-biasing technology to multiple populations statistical inference problem and propose the de-biasing FGL estimators. The corresponding asymptotic property of de-biasing FGL estimators is provided. A simulation study and an application of the diffuse large B-cell lymphoma data show that the proposed test works well in high-dimensional situations.

However, I still have some questions about the manuscript.

1. As mentioned in the paper, the authors need to minimize the negative penalized log-likelihood function (1) and get the algorithm solution as an estimator in both numerical study and real data application. The authors use the ADMM algorithm to execute the estimating process with the classical AIC tuning method. Could the author demonstrate the ADMM algorithm for fused lasso? Will the solutions with different tuning methods affect the accuracy of the hypothesis testing process? Is there any better approach to tune the parameters?

2. In page 2, line 13: ``$\\hat\\Sigma^{-1}$" $\\to$ ``$\\hat\\Sigma^{-1}_{n}$".

3. In page 4, line 2: ``we denote $(A)_{ij}$ its $(i,j)$-entry" $\\to$ ``we denote $(A)_{ij}$ as $(i,j)$-entry of A".

4. In page 4, line 3 and line 18: The authors confused the symbols of the determinant and the cardinality, please distinguish them.

5. In page 11, line 17: what does notation $|A|$ mean? Please clarify the meaning of the symbol.

6. There are too many formulas with numbers. Please remove the numbers of formulas that are not used.

6. PLOS authors have the option to publish the peer review history of their article (what does this mean?). If published, this will include your full peer review and any attached files.

Reviewer #1: No

Reviewer #2: No

---

## [Author Response · Author response to Decision Letter 0]

15 Apr 2024

Thanks for the comments of reviewers and editor. We have uploaded the “respond to reviewers.pdf” to the attachment. Please check the attachment. Thanks again.

---

## [Decision Letter · Decision Letter 1]

9 May 2024

Application of fused graphical lasso to statistical inference for multiple sparse precision matrices

PONE-D-23-38136R1

Dear Dr. Zhang,

We’re pleased to inform you that your manuscript has been judged scientifically suitable for publication and will be formally accepted for publication once it meets all outstanding technical requirements.

Kind regards,

Debo Cheng

Academic Editor

PLOS ONE

Additional Editor Comments (optional):

Both reviewers agreed to accept this manuscript. I agree to recommend acceptance.

Reviewers' comments:

Reviewer's Responses to Questions

**Comments to the Author**

1. If the authors have adequately addressed your comments raised in a previous round of review and you feel that this manuscript is now acceptable for publication, you may indicate that here to bypass the “Comments to the Author” section, enter your conflict of interest statement in the “Confidential to Editor” section, and submit your "Accept" recommendation.

Reviewer #1: (No Response)

Reviewer #2: All comments have been addressed

2. Is the manuscript technically sound, and do the data support the conclusions?

Reviewer #1: (No Response)

Reviewer #2: Yes

3. Has the statistical analysis been performed appropriately and rigorously? 

Reviewer #1: (No Response)

Reviewer #2: Yes

4. Have the authors made all data underlying the findings in their manuscript fully available?

Reviewer #1: (No Response)

Reviewer #2: Yes

5. Is the manuscript presented in an intelligible fashion and written in standard English?

Reviewer #1: (No Response)

Reviewer #2: Yes

6. Review Comments to the Author

Reviewer #1: (No Response)

Reviewer #2: (No Response)

7. PLOS authors have the option to publish the peer review history of their article (what does this mean?). If published, this will include your full peer review and any attached files.

Reviewer #1: No

Reviewer #2: No

---

## [Editor Report · Acceptance letter]

14 May 2024

PONE-D-23-38136R1 

PLOS ONE

Dear Dr. Zhang, 

I'm pleased to inform you that your manuscript has been deemed suitable for publication in PLOS ONE. Congratulations! Your manuscript is now being handed over to our production team.

Kind regards, 

on behalf of

Dr. Debo Cheng 

Academic Editor

PLOS ONE